# High-recall causal discovery for autocorrelated time series with latent confounders

**Andreas Gerhardus**
German Aerospace Center
Institute of Data Science
07745 Jena, Germany
`andreas.gerhardus@dlr.de`

**Jakob Runge**
German Aerospace Center
Institute of Data Science
07745 Jena, Germany
`jakob.runge@dlr.de`

## Abstract

We present a new method for linear and nonlinear, lagged and contemporaneous constraint-based causal discovery from observational time series in the presence of latent confounders. We show that existing causal discovery methods such as FCI and variants suffer from low recall in the autocorrelated time series case and identify low effect size of conditional independence tests as the main reason. Information-theoretical arguments show that effect size can often be increased if causal parents are included in the conditioning sets. To identify parents early on, we suggest an iterative procedure that utilizes novel orientation rules to determine ancestral relationships already during the edge removal phase. We prove that the method is order-independent, and sound and complete in the oracle case. Extensive simulation studies for different numbers of variables, time lags, sample sizes, and further cases demonstrate that our method indeed achieves much higher recall than existing methods for the case of autocorrelated continuous variables while keeping false positives at the desired level. This performance gain grows with stronger autocorrelation. At `github.com/jakobrunge/tigramite` we provide Python code for all methods involved in the simulation studies.

## 1 Introduction

Observational causal discovery [Spirtes et al., 2000, Peters et al., 2017] from time series is a challenge of high relevance to many fields of science and engineering if experimental interventions are infeasible, expensive, or unethical. Causal knowledge of direct and indirect effects, interaction pathways, and time lags can help to understand and model physical systems and to predict the effect of interventions [Pearl, 2000]. Causal graphs can also guide interpretable variable selection for prediction and classification tasks. Causal discovery from time series faces major challenges [Runge et al., 2019a] such as unobserved confounders, high-dimensionality, and nonlinear dependencies, to name a few. Few frameworks can deal with these challenges and we here focus on constraint-based methods pioneered in the seminal works of Spirtes, Glymour, and Zhang [Spirtes et al., 2000, Zhang, 2008]. We demonstrate that existing latent causal discovery methods strongly suffer from low recall in the time series case where identifying lagged and contemporaneous causal links is the goal and autocorrelation is an added, ubiquitous challenge. Our main **theoretical contributions** lie in identifying low effect size as a major reason why current methods fail and in introducing a novel sound, complete, and order-independent causal discovery algorithm that yields strong gains in recall for autocorrelated continuous data. Our **practical contributions** lie in extensive numerical experiments that can serve as a future benchmark and in open-source Python implementations of our and major previous time series causal discovery algorithms. **The paper is structured as follows:** After briefly introducing the problem and existing methods in Sec. 2, we describe our method and theoretical results in Sec. 3.

Section 4 provides numerical experiments followed by a discussion of strengths and weaknesses as well as an outlook in Sec. 6. The paper is accompanied by Supplementary Material (SM).

## 2 Time series causal discovery in the presence of latent confounders

### 2.1 Preliminaries

We consider multivariate time series $\mathbf{V}^j = (V_t^j, V_{t-1}^j, \ldots)$ for $j = 1, \ldots, \tilde{N}$ that follow a stationary discrete-time structural vector-autoregressive process described by the structural causal model (SCM)

$$V_t^j = f_j(pa(V_t^j), \eta_t^j) \quad \text{with } j = 1, \ldots, \tilde{N} . \tag{1}$$

The measurable functions $f_j$ depend non-trivially on all their arguments, the noise variables $\eta_t^j$ are jointly independent, and the sets $pa(V_t^j) \subseteq (\mathbf{V}_t, \mathbf{V}_{t-1}, \ldots, \mathbf{V}_{t-p_{ts}})$ define the causal parents of $V_t^j$. Here, $\mathbf{V}_t = (V_t^1, V_t^2, \ldots)$ and $p_{ts}$ is the order of the time series. Due to stationarity the causal relationship of the pair of variables $(V_{t-\tau}^i, V_t^j)$, where $\tau \geq 0$ is known as lag, is the same as that of all time shifted pairs $(V_{t'-\tau}^i, V_{t'}^j)$. This is why below we always fix one variable at time $t$. We assume that there are no cyclic causal relationships, which as a result of time order restricts the contemporaneous ($\tau = 0$) interactions only. We allow for unobserved variables, i.e., we allow for observing only a subset $\mathbf{X} = \{\mathbf{X}^1, \ldots, \mathbf{X}^N\} \subseteq \mathbf{V} = \{\mathbf{V}^1, \mathbf{V}^2, \ldots\}$ of time series with $N \leq \tilde{N}$. We further assume that there are no selection variables and assume the faithfulness [Spirtes et al., 2000] condition, which states that conditional independence (CI) in the observed distribution $P(\mathbf{V})$ generated by the SCM implies d-separation in the associated time series graph $\mathcal{G}$ over variables $\mathbf{V}$.

We assume the reader is familiar with the Fast Causal Inference (FCI) algorithm [Spirtes et al., 1995, Spirtes et al., 2000, Zhang, 2008] and related graphical terminology, see Secs. S1 and S2 of the SM for a brief overview. Importantly, the MAGs (maximal ancestral graphs) considered in this paper can contain directed ($\rightarrow$) and bidirected ($\leftrightarrow$) edges (interchangeably also called links). The associated PAGs (partial ancestral graphs) may additionally have edges of the type $\circ\!\rightarrow$ and $\circ\!-\!\circ$.

### 2.2 Existing methods

The **tsFCI** algorithm [Entner and Hoyer, 2010] adapts the constraint-based **FCI** algorithm to time series. It uses time order and stationarity to restrict conditioning sets and to apply additional edge orientations. **SVAR-FCI** [Malinsky and Spirtes, 2018] uses stationarity to also infer additional edge removals. There are no assumptions on the functional relationships or on the structure of confounding. **Granger causality** [Granger, 1969] is another common framework for inferring the causal structure of time series. It cannot deal with contemporaneous links (known as instantaneous effects in this context) and may draw wrong conclusions in the presence of latent confounders, see e.g. [Peters et al., 2017] for an overview. The **ANLTSM** method [Chu and Glymour, 2008] restricts contemporaneous interactions to be linear, and latent confounders to be linear and contemporaneous. **TS-LiNGAM** [Hyvärinen et al., 2008] is based on **LiNGAM** [Shimizu et al., 2006] that is rooted in the structural causal model framework [Peters et al., 2017, Spirtes and Zhang, 2016]. It allows for contemporaneous effects, assumes linear interactions with additive non-Gaussian noise, and might fail in the presence of confounding. The **TiMINo** [Peters et al., 2013] method restricts interactions to an identifiable function class or requires an acyclic summary graph. Yet another approach are **Bayesian score-based or hybrid methods** [Chickering, 2002, Tsamardinos et al., 2006]. These often become computationally infeasible in the presence of unobserved variables, see [Jabbari et al., 2017] for a discussion, or make restrictive assumptions about functional dependencies or variable types.

In this paper we follow the constraint-based approach that allows for general functional relationships (both for lagged and contemporaneous interactions), general types of variables (discrete and continuous, univariate and multivariate), and that makes no assumption on the structure of confounding. The price of this generality is that we will not be able to distinguish all members of a Markov equivalence class (although time order and stationarity allow to exclude some members of the equivalence class). Due to its additional use of stationarity we choose SVAR-FCI rather than tsFCI as a baseline and implement the method, restricted to no selection variables, in Python. As a second baseline we implement **SVAR-RFCI**, which is a time series adaption of RFCI along the lines of SVAR-FCI (also restricted to no selection variables). The **RFCI** algorithm [Colombo et al., 2012] is a modification of FCI that does not execute FCI's potentially time consuming second edge removal phase.

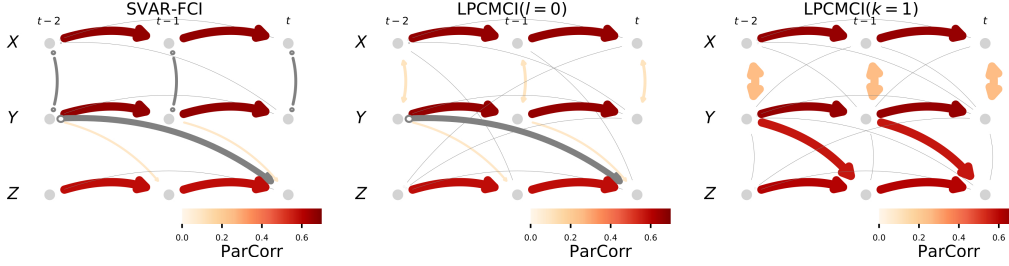

Figure 1: Latent confounder example of the model in eq. (3) (Sec. 4) with linear ground truth links shown for the LPCMCI case (right panel). All auto-coefficients are 0.9, all cross-coefficients are 0.6 (colored links), false links or links with false orientations are grey. True and false adjacency detection rates shown as link width. Detection rates based on 500 realizations run at $\alpha = 0.01$ for $T = 500$.

## 2.3 On maximum time lag, stationarity, soundness, and completeness

In time series causal discovery the assumption of stationarity and the length of the chosen time lag window $t - \tau_{\max} \leq t' \leq t$ play an important role. In the causally sufficient case ($\mathbf{X} = \mathbf{V}$) the causal graph stays the same for all $\tau_{\max} \geq p_{ts}$. Not so in the latent case: Let $\mathcal{M}(\mathcal{G})^{\tau_{\max}}$ be the MAG obtained by marginalizing over all unobserved variables and also all generally observed variables at times $t' < t - \tau_{\max}$. Then, increasing the considered time lag window by increasing $\tau_{\max}$ may result in the removal of edges that are fully contained in the original window, even in the case of perfect statistical decisions. In other words, $\mathcal{M}(\mathcal{G})^{\tau_{\max,1}}$ with $\tau_{\max,1} < \tau_{\max,2}$ need not be a subgraph of $\mathcal{M}(\mathcal{G})^{\tau_{\max,2}}$. Hence, $\tau_{\max}$ may be regarded more as an analysis choice than as a tunable parameter. For the same reason stationarity also affects the definition of MAGs and PAGs that are being estimated. For example, SVAR-FCI uses stationarity to also remove edges whose separating set extends beyond the chosen time lag window. It does, therefore, in general not determine a PAG of $\mathcal{M}(\mathcal{G})^{\tau_{\max}}$. To formalize this let $i)$ $\mathcal{M}(\mathcal{G})^{\tau_{\max}}_{statA}$ be the MAG obtained from $\mathcal{M}(\mathcal{G})^{\tau_{\max}}$ by enforcing repeating adjacencies, let $ii)$ $\mathcal{P}(\mathcal{G})^{\tau_{\max}}_{statA}$ be the maximally informative PAG for the Markov equivalence class of $\mathcal{M}(\mathcal{G})^{\tau_{\max}}_{statA}$, which can be obtained from running the FCI orientation rules on $\mathcal{M}(\mathcal{G})^{\tau_{\max}}_{statA}$, and let $iii)$ $\mathcal{P}(\mathcal{G})^{\tau_{\max}}_{statAO}$ be the PAG obtained when additionally enforcing time order and repeating orientations at each step of applying the orientation rules. Note that $\mathcal{P}(\mathcal{G})^{\tau_{\max}}_{statAO}$ may have fewer circle marks, i.e., may be more informative than $\mathcal{P}(\mathcal{G})^{\tau_{\max}}_{statA}$. Our aim is to estimate $\mathcal{P}(\mathcal{G})^{\tau_{\max}}_{statAO}$. We say an algorithm is *sound* if it returns a PAG for $\mathcal{M}(\mathcal{G})^{\tau_{\max}}_{statA}$, and *complete* if it returns $\mathcal{P}(\mathcal{G})^{\tau_{\max}}_{statAO}$. Below we write $\mathcal{M}(\mathcal{G}) = \mathcal{M}(\mathcal{G})^{\tau_{\max}}_{statA}$ and $\mathcal{P}(\mathcal{G}) = \mathcal{P}(\mathcal{G})^{\tau_{\max}}_{statAO}$ for simplicity.

## 2.4 Motivational example

We illustrate the challenge posed by unobserved variables with the example of Fig. 1. SVAR-FCI with the partial correlation (ParCorr) CI test correctly identifies the auto-links but misses the true lagged link $Y_{t-1} \rightarrow Z_t$ and returns a false link $Y_{t-2} \rightarrow Z_t$ instead. In most realizations the algorithm fails to detect the contemporaneous adjacency $X_t \leftrightarrow Y_t$ and, if detected, fails to orient it as bidirected. The reason are wrong CI tests in its edge removal and orientation phases. When it iterates through conditioning sets of cardinality $p = 0$ in the **edge removal phase**, the correlation $\rho(X_t; Y_t)$ is non-significant in many realizations since the high autocorrelation of both $X$ and $Y$ increases their variance and *decreases* their signal-to-noise ratio (the common signal due to the latent confounder). Further, for $p = 1$ also the lagged correlation $\rho(Y_{t-1}; Z_t | Y_{t-2})$ often is non-significant and the true link $Y_{t-1} \rightarrow Z_t$ gets removed. Here conditioning away the autocorrelation of $Y_{t-1}$ decreases the signal while the noise level in $Z_t$ is still high due to $Z$'s autocorrelation. This false negative has implications for further CI tests since $Y_{t-1}$ won't be used in subsequent conditioning sets: The path $Y_{t-2} \rightarrow Y_{t-1} \rightarrow Z_t$ can then not be blocked anymore and the false positive $Y_{t-2} \rightarrow Z_t$ remains even after the next removal phase. In the **orientation phase** of SVAR-FCI rule $\mathcal{R}1$ yields tails for all auto-links. Even if the link $X_t \circ\!\!-\!\!\circ Y_t$ is detected, it is in most cases not oriented correctly. The reason again lies in wrong CI tests: In principle the collider rule $\mathcal{R}0$ should identify $X_t \leftrightarrow Y_t$ since the middle node of the triple $X_{t-1} \circ\!\!\rightarrow X_t \circ\!\!-\!\!\circ Y_t$ does *not* lie in the separating set of $X_{t-1}$ and $Y_t$ (and similarly for $X$ and $Y$ swapped). In practice $\mathcal{R}0$ is implemented with the majority rule [Colombo and Maathuis, 2014] to avoid order-dependence, which involves further CI test given subsets of the adjacencies of $X_{t-1}$ and $Y_t$. SVAR-FCI here finds independence given $Y_{t-1}$ (correct) but also given $X_t$ (wrong, due

to autocorrelation). Since the middle node $X_t$ is in exactly half of the separating sets, the triple is marked as ambiguous and left unoriented. The same applies when $X$ and $Y$ are swapped.

Autocorrelation is only one manifestation of a more general problem we observe here: Low signal-to-noise ratio due to an 'unfortunate' choice of conditioning sets that leads to *low effect size* (here partial correlation) and, hence, low statistical power of CI tests. Wrong CI tests then lead to missing links, and these in turn to false positives and wrong orientations. In the following we analyze effect size more theoretically and suggest a general idea to overcome this issue.

## 3 Latent PCMCI

### 3.1 Effect size in causal discovery

The detection power of a true link $X_{t-\tau}^i \ast\!\!\rightarrow X_t^j$, where below we write $A = X_{t-\tau}^i$ and $B = X_t^j$ to emphasize that the discussion also applies to the non-time series case, quantifies the probability of the link not being erroneously removed due to a wrong CI test. It depends on $i$) the sample size (usually fixed), $ii$) the CI tests' significance level $\alpha$ (fixed by the researcher as the desired false positives level), $iii$) the CI tests' estimation dimensions (kept at a minimum by SVAR-FCI's design to preferentially test small conditioning sets), and $iv$) the effect size. We here define effect size as the minimum of the CI test statistic values $I(A; B|\mathcal{S})$ taken over all conditioning sets $\mathcal{S}$ that are being tested (for fixed $A$ and $B$). As observed in the motivating example, this minimum can become very small and hence lead to low detection power. The central idea of **our proposed method Latent PCMCI (LPCMCI)** is to increase effect size by $a$) restricting the conditioning sets $\mathcal{S}$ that need to be tested in order to remove all wrong links, and by $b$) extending those sets $\mathcal{S}$ that do need to be tested with so called *default conditions* $\mathcal{S}_{def}$ that increase the CI test statistic values and at the same time do not induce spurious dependencies. Regarding $a$), Lemma S5 proves that it is sufficient to only consider conditioning sets that consist of ancestors of $A$ or $B$ only. Regarding $b$), and well-fitting with $a$), Lemma S4 proves that no spurious dependencies are introduced if $\mathcal{S}_{def}$ consist of ancestors of $A$ or $B$ only. Further, the following theorem shows that taking $\mathcal{S}_{def}$ as the union of the parents of $A$ and $B$ (without $A$ and $B$ themselves) improves the effect size of LPCMCI over that of SVAR-FCI. This generalizes the *momentary conditional independence* (MCI) idea that underlies the PCMCI and PCMCI$^+$ algorithms [Runge et al., 2019b, Runge, 2020] to causal discovery with latent confounders. We state the theorem in an information theoretic framework, where $I$ denotes (conditional) mutual information and $\mathcal{I}(A; B; C|D) \equiv I(A; B|D) - I(A; B|C \cup D)$ the *interaction information*.

**Theorem 1** (LPCMCI effect size). *Let $A\ast\!\!\rightarrow B$ (with $A = X_{t-\tau}^i$ and $B = X_t^j$) be a link ($\rightarrow$ or $\leftrightarrow$) in $\mathcal{M}(\mathcal{G})$. Consider the default conditions $\mathcal{S}_{def} = pa(\{A, B\}, \mathcal{M}(\mathcal{G})) \setminus \{A, B\}$ and denote $\mathbf{X}^* = \mathbf{X} \setminus \mathcal{S}_{def}$. Let $\mathbf{S} = \arg\min_{\mathcal{S} \subseteq \mathbf{X}^* \setminus \{A, B\}} I(A; B|\mathcal{S} \cup \mathcal{S}_{def})$ be the set of sets that define LPCMCI's effect size. If $i$) there is $\mathcal{S}^* \in \mathbf{S}$ with $\mathcal{S}^* \subseteq adj(A, \mathcal{M}(\mathcal{G})) \setminus \mathcal{S}_{def}$ or $\mathcal{S}^* \subseteq adj(B, \mathcal{M}(\mathcal{G})) \setminus \mathcal{S}_{def}$ and $ii$) there is a proper subset $\mathcal{Q} \subset \mathcal{S}_{def}$ such that $\mathcal{I}(A; B; \mathcal{S}_{def} \setminus \mathcal{Q}|\mathcal{S}^* \cup \mathcal{Q}) < 0$, then*

$$\min_{\mathcal{S} \subseteq \mathbf{X}^* \setminus \{A, B\}} I(A; B|\mathcal{S} \cup \mathcal{S}_{def}) > \min_{\tilde{\mathcal{S}} \subseteq \mathbf{X} \setminus \{A, B\}} I(A; B|\tilde{\mathcal{S}}). \qquad (2)$$

*If the assumptions are not fulfilled, then (trivially) "$\geq$" holds in eq. (2).*

The second assumption only requires that *any* subset $\mathcal{S}_{def} \setminus Q$ of the parents contains information that increases the information between $A$ and $B$. A sufficient condition for this is detailed in Corollary S1.

These considerations lead to two design principles behind LPCMCI: First, when testing for conditional independence of $A$ and $B$, discard conditioning sets that contain known non-ancestors of $A$ and $B$. Second, use known parents of $A$ and $B$ as default conditions. Unless the higher effect size is overly counteracted by the increased estimation dimension (due to conditioning sets of higher cardinality), this leads to higher detection power and hence higher recall of true links. While we do not claim that our choice of default conditions as further detailed in Sec. 3.4 is optimal, our numerical experiments in Sec. 4 and the SM indicate strong increases in recall for the case of continuous variables with autocorrelation. In [Runge et al., 2019b, Runge, 2020] it is discussed that, in addition to higher effect size, conditioning on the parents of both $A$ and $B$ also leads to better calibrated tests which in turn avoids inflated false positives. Another benefit is that fewer conditioning sets need to be tested, which is also the motivation for a default conditioning on known parents in [Lee and Honavar, 2020].

The above design principles are only useful if some (non-)ancestorships are known before all CI test have been completed. LPCMCI achieves this by entangling the edge removal and edge orientation

phases, i.e., by learning ancestral relations before having removed all wrong links. For this purpose we below develop novel orientation rules. These are not necessary in the causally sufficienct setting considered by PCMCI$^+$ [Runge, 2020] because there the default conditions need not be limited to ancestors of $A$ or $B$ (although PCMCI$^+$ tries to keep the number of default conditions low). While not considered here, background knowledge about (non-)ancestorships can easily be incorporated.

## 3.2 Introducing middle marks and LPCMCI-PAGs

To facilitate early orientation of edges we give an unambiguous causal interpretation to the graph at every step of the algorithm. This is achieved by augmenting edges with *middle marks*. Using generic variable names $A$, $B$, and $C$ indicates that the discussion also applies to the non-time series case.

Middle marks are denoted above the link symbol and can be '?', 'L', 'R', '!', or '' (empty). The 'L' ('R') on $A*\overset{L}{-}*B$ ($A*\overset{R}{-}*B$) asserts that if $A < B$ ($B < A$) then $B \notin an(A, \mathcal{G})$ or there is no $\mathcal{S} \subseteq pa(A, \mathcal{M}(\mathcal{G}))$ that m-separates $A$ and $B$ in $\mathcal{M}(\mathcal{G})$. Here $<$ is any total order on the set of variables. Its choice is arbitrary and does not influence the causal information content, the sole purpose being to disambiguate $A*\overset{L}{-}*B$ from $A*\overset{R}{-}*B$. Moreover, '$*$' is a wildcard that may stand for all three edge marks (tail, head, circle) that appear in PAGs. Further, the '!' on $A*\overset{!}{-}*B$ asserts that both $A*\overset{L}{-}*B$ and $A*\overset{R}{-}*B$ are true, and the empty middle mark on $A*\!\!-\!\!*B$ says that $A \in adj(B, \mathcal{M}(\mathcal{G}))$. Lastly, the '?' on $A*\overset{?}{-}*B$ doesn't promise anything. Non-circle edge marks (here potentially hidden by the '$*$' symbol) still convey their standard meaning of ancestorship and non-ancestorship, and the absence of an edge between $A$ and $B$ still asserts that $A \notin adj(B, \mathcal{M}(\mathcal{G}))$. We call a PAG $\mathcal{C}(\mathcal{G})$ whose edges are extended with middle marks a LPCMCI-PAG for $\mathcal{M}(\mathcal{G})$, see Sec. S3 in the SM for a more formal definition. The '$*$' symbol is also used as a wildcard for the five middle marks.

Note that we are *not* changing the quantity we are trying to estimate, this is still the PAG $\mathcal{P}(\mathcal{G})$ as explained in Sec. 2.3. The notion of LPCMCI-PAGs is used in intermediate steps of LPCMCI and has two advantages. First, $A*\!\!-\!\!*B$ is reserved for $A \in adj(B, \mathcal{M}(\mathcal{G}))$ and thus has an unambiguous meaning at every point of the algorithm, unlike for (SVAR-)FCI and (SVAR-)RFCI. In fact, even if LPCMCI is interrupted at any arbitrary point it still yields a graph with unambiguous and sound causal interpretation. Second, middle marks carry fine-grained causal information that allows to determine definite adjacencies early on:

**Lemma 1** (Ancestor-parent-rule). *In LPCMCI-PAG $\mathcal{C}(\mathcal{G})$ one may replace 1.) $A\overset{!}{\rightarrow}B$ by $A\rightarrow B$, 2.) $A\overset{L}{\rightarrow}B$ for $A > B$ by $A\rightarrow B$, and 3.) $A\overset{R}{\rightarrow}B$ for $A < B$ by $A\rightarrow B$.*

When LPCMCI has converged all middle marks are empty and hence $\mathcal{C}(\mathcal{G})$ is a PAG. We choose a total order consistent with time order, namely $X^i_{t-\tau} < X^j_t$ iff $\tau > 0$ or $\tau = 0$ and $i < j$. Lagged links can then be initialized with edges $\circ\overset{L}{\rightarrow}$ (contemporaneous links as $\circ\overset{?}{-}\circ$).

## 3.3 Orientations rules for LPCMCI-PAGs

We now discuss rules for edge orientation in LPCMCI-PAGs. For this we need a definition:

**Definition 1** (Weakly minimal separating sets). *In MAG $\mathcal{M}(\mathcal{G})$ let $A$ and $B$ be m-separated by $\mathcal{S}$. The set $\mathcal{S}$ is a weakly minimal separating set of $A$ and $B$ if i) it decomposes as $\mathcal{S} = \mathcal{S}_1 \dot{\cup} \mathcal{S}_2$ with $\mathcal{S}_1 \subseteq an(\{A, B\}, \mathcal{M}(\mathcal{G}))$ such that ii) if $\mathcal{S}' = \mathcal{S}_1 \dot{\cup} \mathcal{S}_2'$ with $\mathcal{S}_2' \subseteq \mathcal{S}_2$ m-separates $A$ and $B$ then $\mathcal{S}_2' = \mathcal{S}_2$. The pair $(\mathcal{S}_1, \mathcal{S}_2)$ is called a weakly minimal decomposition of $\mathcal{S}$.*

This generalizes the notion of minimal separating sets, for which additionally $\mathcal{S}_1 = \emptyset$. Since LPCMCI is designed to extend conditioning sets by known ancestors, the separating sets it finds are in general not minimal. However, they are still weakly minimal. The following Lemma, a generalization of the unshielded triple rule [Colombo et al., 2012], is central to orientations in LPCMCI-PAGs:

**Lemma 2** (Strong unshielded triple rule). *Let $A*\overset{?}{-}*B*\overset{?}{-}*C$ be an unshielded triple in LPCMCI-PAG $\mathcal{C}(\mathcal{G})$ and $\mathcal{S}_{AC}$ the separating set of $A$ and $C$. 1.) If i) $B \in \mathcal{S}_{AC}$ and ii) $\mathcal{S}_{AC}$ is weakly minimal, then $B \in an(\{A, C\}, \mathcal{G})$. 2.) Let $\mathcal{T}_{AB} \subseteq an(\{A, B\}, \mathcal{M}(\mathcal{G}))$ and $\mathcal{T}_{CB} \subseteq an(\{C, B\}, \mathcal{M}(\mathcal{G}))$ be arbitrary. If i) $B \notin \mathcal{S}_{AC}$, ii) $A$ and $B$ are not m-separated by $\mathcal{S}_{AC} \cup \mathcal{T}_{AB} \setminus \{A, B\}$, iii) $C$ and $B$ are not m-separated by $\mathcal{S}_{AC} \cup \mathcal{T}_{CB} \setminus \{C, B\}$, then $B \notin an(\{A, C\}, \mathcal{G})$. The conditioning sets in ii) and iii) may be intersected with the past and present of the later variable.*

Part 2.) of this Lemma generalizes the FCI collider rule $\mathcal{R}0$ to rule $\mathcal{R}0'$ (of which there are several variations when restricting to particular middle marks), and part 1.) generalizes $\mathcal{R}1$ to $\mathcal{R}1'$. Rules $\mathcal{R}2$

and $\mathcal{R}8$ generalize trivially to triangles in $\mathcal{C}(\mathcal{G})$ with arbitrary middle marks, giving rise to $\mathcal{R}2'$ and $\mathcal{R}8'$. Rules $\mathcal{R}3$, $\mathcal{R}9$ and $\mathcal{R}10$ are generalized to $\mathcal{R}3'$, $\mathcal{R}9'$ and $\mathcal{R}10'$ by adding the requirement that the middle variables of certain unshielded colliders are in the separating set of the two outer variables, and that these separating sets are weakly minimal. Since there are no selection variables, rules $\mathcal{R}5$, $\mathcal{R}6$ and $\mathcal{R}7$ are not applicable. Rule $\mathcal{R}4'$ generalizes the discriminating path rule [Colombo et al., 2012] of RFCI. These rules are complemented by the replacements specified in Lemma 1 and a rule for updating middle marks. Precise formulations of all rules are given in Sec. S4 of the SM.

We stress that these rules are applicable at every point of the algorithm and that they may be executed in any order. This is different from the (SVAR-)FCI orientation phase which requires that prior to orientation a PAG has been found. Also (SVAR-)RFCI orients links only once an RFCI-PAG has been determined, and both (SVAR-)FCI and (SVAR-)RFCI require that all colliders are oriented before applying their other orientation rules.

### 3.4  The LPCMCI algorithm

**LPCMCI** is a constraint-based causal discovery algorithm that utilizes the findings of Sec. 3.1 to increase the effect size of CI tests. High-level pseudocode is given in **Algorithm 1**. After initializing $\mathcal{C}(\mathcal{G})$ as a complete graph, the algorithm enters its *preliminary phase* in lines 2 to 4. This involves calls to **Algorithm S2** (pseudocode in Sec. S5 of the SM), which removes many (but in general not all) false links and, while doing so, repeatedly applies the orientation rules introduced in the previous section. These rules identify a subset of the (non-)ancestorships in $\mathcal{G}$ and accordingly mark them by heads or tails on edges in $\mathcal{C}(\mathcal{G})$. This information is then used as prescribed by the two design principles of LPCMCI that were explained in Sec. 3.1: The non-ancestorships further constrain the conditioning sets $\mathcal{S}$ of subsequent CI tests, the ancestorships are used to extend these sets to $\mathcal{S} \cup \mathcal{S}_{def}$ where $\mathcal{S}_{def} = pa(\{X^i_{t-\tau}, X^j_t\}, \mathcal{C}(\mathcal{G}))$ are the by then known parents of those variables whose independence is being tested. All parentships marked in $\mathcal{C}(\mathcal{G})$ after line 3 are remembered and carried over to an elsewise re-initialized $\mathcal{C}(\mathcal{G})$ before the next application of Alg. S2. Conditioning sets can then be extended with known parents already from the beginning. The purpose of this iterative process is to determine an accurate subset of the parentships in $\mathcal{G}$. These are then passed on to the *final phase* in lines 5 - 6, which starts with one final application of Alg. S2. At this point there may still be false links because Alg. S2 may fail to remove a false link between variables $X^i_{t-\tau}$ and $X^j_t$ if neither of the two is an ancestor of the other. This is the purpose of **Algorithm S3** (pseudocode in Sec. S5 of the SM) that is called in line 6, which thus plays a similar role as the second removal phase in (SVAR-)FCI. Algorithm S3 repeatedly applies orientation rules and uses identified (non-)ancestorships in the same way as Alg. S2. As stated in the following theorems, LPCMCI will then have found the PAG $\mathcal{P}(\mathcal{G})$. Moreover, its output does not depend on the order of the $N$ time series variables $X^j$. The number $k$ of iterations in the preliminary phase is a hyperparameter and we write LPCMCI($k = k_0$) when specifying $k = k_0$. Stationarity is enforced at every step of the algorithm, i.e., whenever an edge is removed or oriented all equivalent time shifted edges (called 'homologous' in [Entner and Hoyer, 2010]) are removed too and oriented in the same way.

---

**Algorithm 1** LPCMCI

---

**Require:** Time series dataset $\mathbf{X} = \{\mathbf{X}^1, \ldots, \mathbf{X}^N\}$, maximal considered time lag $\tau_{\max}$, significance level $\alpha$, CI test CI$(X, Y, \mathcal{S})$, non-negative integer $k$
1: Initialize $\mathcal{C}(\mathcal{G})$ as complete graph with $X^i_{t-\tau} \circ\!\!\overset{L}{\rightarrow} X^j_t$ ($0 < \tau \leq \tau_{\max}$) and $X^i_{t-\tau} \circ\!\!\overset{?}{\rightarrow}\!\circ X^j_t$ ($\tau = 0$)
2: **for** $0 \leq l \leq k - 1$ **do**
3:     Remove edges and apply orientations using Algorithm S2
4:     Repeat line 1, orient edges as $X^i_{t-\tau} \overset{?}{\rightarrow} X^j_t$ if $X^i_{t-\tau} \overset{*}{\rightarrow} X^j_t$ was in $\mathcal{C}(\mathcal{G})$ after line 3
5: Remove edges and apply orientations using Algorithm S2
6: Remove edges and apply orientations using Algorithm S3
7: **return** PAG $\mathcal{C}(\mathcal{G}) = \mathcal{P}(\mathcal{G}) = \mathcal{P}(\mathcal{G})^{\tau_{\max}}_{statAO}$

---

**Theorem 2** (LPCMCI is sound and complete). *Assume that there is a process as in eq. (1) without causal cycles, which generates a distribution $P$ that is faithful to its time series graph $\mathcal{G}$. Further assume that there are no selection variables, and that we are given perfect statistical decisions about CI of observed variables in $P$. Then LPCMCI is sound and complete, i.e., it returns the PAG $\mathcal{P}(\mathcal{G})$.*

**Theorem 3** (LPCMCI is order-independent). *The output of LPCMCI does not depend on the order of the $N$ time series variables $X^j$ (the $j$-indices may be permuted).*

### 3.5 Back to the motivational example in Fig. 1

The **first iteration** ($l = 0$) of LPCMCI also misses the links $Y_{t-1} \to Z_t$ and finds $X_t \ast\!\!-\!\!\ast Y_t$ in only few realizations (we here suppress middle marks for simpler notation), but orientations are already improved as compared to SVAR-FCI. Rule $\mathcal{R}1'$ applied after $p = 1$ orients the auto-links $X_{t-1} \to X_t$ and $Y_{t-1} \to Y_t$. This leads to the parents sets $pa(X_t, \mathcal{C}(\mathcal{G})) = \{X_{t-1}\}$ and $pa(Y_t, \mathcal{C}(\mathcal{G})) = \{Y_{t-1}\}$, which are then used as default conditions in subsequent CI tests. This is relevant for orientation rule $\mathcal{R}0'$ that tests whether the middle node of the unshielded triple $X_{t-1} \circ\!\!\to X_t \circ\!\!-\!\!\circ Y_t$ does *not* lie in the separating set of $X_{t-1}$ and $Y_t$. Due to the extra conditions the relevant partial correlation $\rho(X_{t-1}; Y_t | X_t, X_{t-2}, Y_{t-1})$ now correctly turns out significant. This identifies $X_t$ as collider and (since the same applies with $X$ and $Y$ swapped) the bidirected edge $X_t \leftrightarrow Y_t$ is correctly found. The **next iteration** ($l = 1$) then uses the parents obtained in the $l = 0$ iteration, here the autodependencies plus the (false) link $Y_{t-2} \to Z_t$, as default conditions already from the beginning for $p = 0$. While the correlation $\rho(X_t; Y_t)$ used by SVAR-FCI is often non-significant, the partial correlation $\rho(X_t; Y_t | X_{t-1}, Y_{t-1})$ is significant since the autocorrelation noise was removed and effect size increased (indicated as link color in Fig. 1) in accord with Theorem 1. Also the lagged link is correctly detected because $\rho(Y_{t-1}; Z_t | Y_{t-2}, Z_{t-1})$ is larger than $\rho(Y_{t-1}; Z_t | Y_{t-2})$. The false link $Y_{t-2} \to Z_t$ is now removed since the separating node $Y_{t-1}$ was retained. This wrong parentship is then also not used for default conditioning anymore. Orientations of bidirected links are facilitated as before and $Y_{t-1} \to Z_t$ is oriented by rule $\mathcal{R}1'$.

## 4 Numerical experiments

We here compare LPCMCI to the SVAR-FCI and SVAR-RFCI baselines with CI tests based on linear partial correlation (ParCorr), for an overview of further experiments presented in the SM see the end of this section. To limit runtime we constrain the cardinality of conditioning sets to 3 in the second removal phase of SVAR-FCI and in Alg. S3 of LPCMCI (excluding the default conditions $\mathcal{S}_{def}$, i.e., $|\mathcal{S}| \leq 3$ but $|\mathcal{S} \cup \mathcal{S}_{def}| > 3$ is allowed). We generate datasets with this variant of the SCM in eq. (1):

$$V_t^j = a_j V_{t-1}^j + \sum_i c_i f_i(V_{t-\tau_i}^i) + \eta_t^j \quad \text{for} \quad j \in \{1, \dots, \tilde{N}\} \tag{3}$$

Autocorrelations $a_j$ are drawn uniformly from $[\max(0, a - 0.3), a]$ for some $a$ as indicated in Fig. 2. For each model we in addition randomly choose $L = \tilde{N}$ linear (i.e., $f_i = id$) cross-links with the corresponding non-zero coefficients $c_i$ drawn uniformly from $\pm[0.2, 0.8]$. 30% of these links are contemporaneous (i.e., $\tau_i = 0$), the remaining $\tau_i$ are drawn from $[1, p_{ts} = 3]$. The noises $\eta^j \sim \mathcal{N}(0, \sigma_j^2)$ are *iid* with $\sigma_j$ drawn from $[0.5, 2]$. We only consider stationary models. From the $\tilde{N}$ variables of each model we randomly choose $N = \lceil (1 - \lambda)\tilde{N} \rceil$ for $\lambda = 0.3$ as observed. As discussed in Sec. 2.3, the true PAG $\mathcal{P}(\mathcal{G})$ of each model depends on $\tau_{\max}$. In Fig. 2 we show the relative average numbers of directed, bidirected, and (partially) unoriented links. For performance evaluation true positive (= recall) and false positive rates for adjacencies are distinguished between lagged cross-links ($i \neq j$), contemporaneous, and autodependency links. False positives instead of precision are shown to investigate whether methods can control these below the $\alpha$-level. Orientation performance is evaluated based on edgemark recall and precision. In Fig. 2 we also show the average of minimum absolute ParCorr values as an estimate of effect size and the average maximum cardinality of all tested conditioning sets. All metrics are computed across all estimated graphs from 500 realizations of the model in eq. (3) at time series length $T$. The average and 90% range of runtime estimates were evaluated on Intel Xeon Platinum 8260.

In **Fig. 2A** we show LPCMCI for $k = 0, \dots, 4$ against increasing autocorrelation $a$. Note that $a = 0$ implies a different true PAG than $a > 0$. The largest gain, both in recall and precision, comes already from $k = 0$ to $k = 1$. For higher $k$ LPCMCI maintains false positive control and orientation precision, and improves recall before converging at $k = 4$. The gain in recall is largely attributable to improved effect size. On the downside, larger $k$ increase cardinality (estimation dimension) and runtime. However, the runtime increase is only marginal because later $l$-steps converge faster and the implementation caches CI test results. **Fig. 2B** shows a comparison of LPCMCI with SVAR-FCI and SVAR-RFCI against autocorrelation, depicting LPCMCI for $k = 0$ and $k = 4$. Already

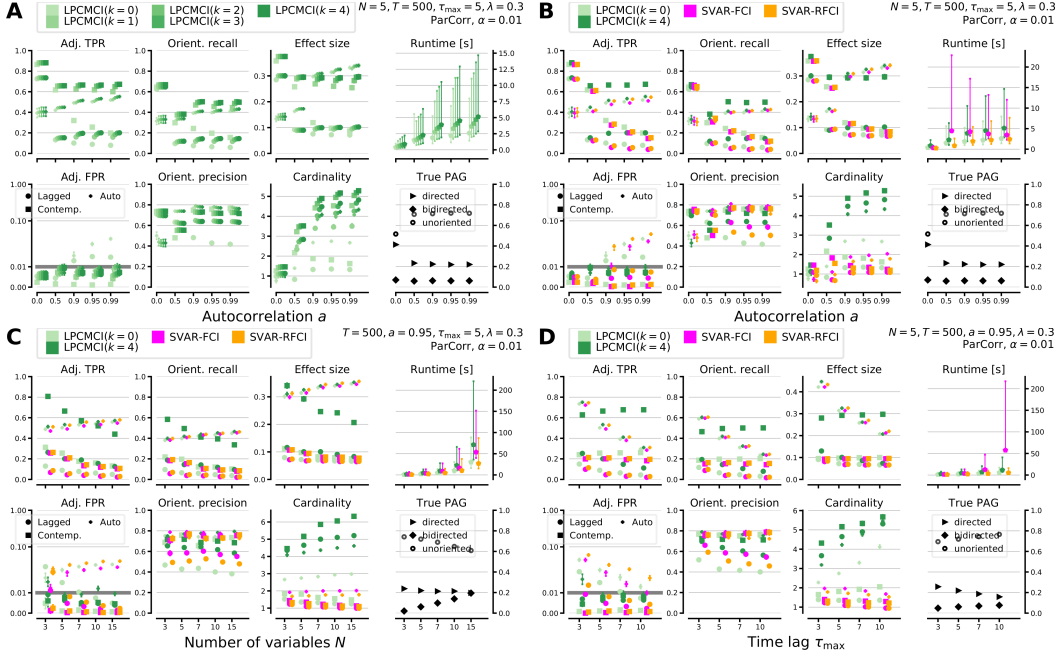

Figure 2: Results of numerical experiments for (**A**) LPCMCI($k$) for different $k$, LPCMCI compared to SVAR-FCI and SVAR-RFCI for (**B**) varying autocorrelation, for (**C**) number of variables $N$, and for (**D**) maximum time lag $\tau_{\max}$ (other parameters indicated in upper right of each panel).

LPCMCI($k = 0$) has higher adjacency and orientation recall than SVAR-FCI and SVAR-RFCI for increasing autocorrelation while they are on par for $a = 0$. This comes at the price of precision, especially lagged orientation precision. LPCMCI($k = 4$) has more than 0.4 higher contemporaneous orientation recall and still 0.1 higher lagged orientation recall than SVAR-FCI and SVAR-RFCI. Lagged precision is higher for high autocorrelation and contemporaneous precision is slightly lower. LPCMCI($k = 4$) maintains high recall for increasing autocorrelation $a \geq 0.5$ while SVAR-FCI and SVAR-RFCI's recall sharply drops. These results can be explained by improved effect size while the increased cardinality ($\approx 5$) of separating sets is still moderate compared to the sample size $T = 500$. LPCMCI($k = 0$) has similar low runtime as SVAR-RFCI, for LPCMCI($k = 4$) it is comparable to that of SVAR-FCI. In **Fig. 2C** we show results for different numbers of variables $N$. As expected, all methods have decreasing adjacency and orientation recall for higher $N$, but LPCMCI starts at a much higher level. For $N = 3$ both SVAR-FCI and SVAR-RFCI cannot control false positives for lagged links while for larger $N$ false positives become controlled. The reason is the interplay of ill-calibrated CI tests for smaller $N$ due to autocorrelation (inflating false positives) with sequential testing for larger $N$ (reducing false positives), as has been discussed in [Runge et al., 2019b, Runge, 2020] for the similar PC algorithm [Spirtes and Glymour, 1991]. LPCMCI better controls false positives here, its decreasing recall can be explained by decreasing effect size and increasing cardinality. Runtime becomes slightly larger than that of SVAR-FCI for larger $N$. **Fig. 2D** shows results for different maximum time lags $\tau_{\max}$. Note that these imply different true PAGs, especially since further lagged links appear for larger $\tau_{\max}$. All methods show a decrease in lagged recall and precision, whereas contemporaneous recall and precision stay almost constant. For SVAR-FCI there is an explosion of runtime for higher $\tau_{\max}$ due to excessive searches of separating sets in its second removal phase. In LPCMCI this is partially overcome since the sets that need to be searched through are more restricted.

In Sec. S9 in the SM we present further numerical experiments. This includes more combinations of model parameters $N$, $a$, $\lambda$, $T$, **nonlinear** models together with the nonparametric GPDC CI test [Runge et al., 2019b], and a comparison to a **residualization** approach. In these cases the results are largely comparable to those above regarding relative performances. For **non-time series** models we find that, although all findings of Secs. 3.1 through 3.4 still apply, LPCMCI($k$) is on par with the baselines for $k = 0$ while it shows inflated false positives for $k = 4$. Similarly, for models of **discrete variables** together with a $G$-test of conditional independence LPCMCI($k$) performs comparable to

the baselines for $k = 0$ and gets worse with increasing $k$. A more detailed analysis of LPCMCI's performance in these two cases, non-time series and discrete models, is subject to future research.

# 5    Application to real data

We here discuss an application of LPCMCI to average daily discharges of rivers in the upper Danube basin, measurements of which are made available by the Bavarian Environmental Agency at https://www.gkd.bayern.de. We consider measurements from the Iller at Kempten ($X$), the Danube at Dillingen ($Y$), and the Isar at Lenggries ($Z$). While the Iller discharges into the Danube upstream of Dillingen with the water from Kempten reaching Dillingen within about a day, the Isar reaches the Danube downstream of Dillingen. We thus expect a contemporaneous link $X_t{\rightarrow}Y_t$ and no direct causal relationships between the pairs $X, Z$ and $Y, Z$. Since all variables may be confounded by rainfall or other weather conditions, this choice allows to test the ability of detecting and distinguishing directed and bidirected links. To keep the sample size comparable with those in the simulation studies we restrict to the records of the past three years (2017-2019). We set $\tau_{\max} = 2$ and apply LPCMCI($k$) for $k = 0, \dots, 4$ and $\alpha = 0.01$ with ParCorr CI tests. Restricting the discussion to contemporaneous links, LPCMCI correctly finds $X_t{\rightarrow}Y_t$ for $k = 1, \dots, 4$ and for $k = 0$ wrongly finds $X_t{\leftrightarrow}Y_t$. For all $k$ it infers the bidirected link $X_t{\leftrightarrow}Z_t$, which is plausible due to confounding by weather. For $k = 3, 4$ LPCMCI wrongly finds the directed link $Z_t{\rightarrow}Y_t$, which should either be absent or bidirected. The results are similar for $\alpha = 0.05$, with the difference that LPCMCI then always correctly finds $X_t{\rightarrow}Y_t$ but wrongly infers $Z_t{\rightarrow}Y_t$ also for $k = 1, 2$. In comparison, SVAR-FCI with ParCorr CI tests finds the contemporaneous adjacencies $Y_t{\circ}{-}{\circ}X_t{\circ}{-}{\circ}Z_t$ for $\alpha = 0.01, 0.03, 0.05, 0.08, 0.1, 0.3, 0.5$ and $Y_t{\leftarrow}{\circ}X_t{\circ}{-}{\circ}Z_t$ for $\alpha = 0.8$. The estimated PAGs are shown in Sec. S10 of the SM.

We note that since the discharge values show extreme events caused by heavy rainfall, the assumption of stationarity is expected to be violated. For other analyses of the dataset of average daily discharges see [Asadi et al., 2015, Engelke and Hitz, 2020, Mhalla et al., 2020, Gnecco et al., 2020]. More detailed applications to and analyses of LPCMCI on real data are subject to future research.

# 6    Discussion and future work

Major **strengths** of LPCMCI lie in its significantly improved recall as compared to the SVAR-FCI and SVAR-RFCI baselines for autocorrelated continuous variables, which grows with autocorrelation and is particularly strong for contemporaneous links. At the same time LPCMCI (for $k > 0$) has better calibrated CI test leading to better false positive control than the baselines. We cannot prove false positive control, but are not aware of any such proof for other constraint-based algorithms in the challenging latent, nonlinear, autocorrelated setting considered here. A general **weakness**, which also applies to (SVAR-)FCI and (SVAR-)RFCI, is the faithfulness assumption. If violated in practice this may lead to wrong conclusions. We did not attempt to only assume the weaker form of adjacency-faithfulness [Ramsey et al., 2006], which to our knowledge is however generally an open problem in the causally insufficient case. Moreover, like all constraint-based methods, our method cannot distinguish all members of Markov equivalence classes like methods based on the SCM framework such as e.g. TS-LiNGAM [Hyvärinen et al., 2008] and TiMINo [Peters et al., 2013] do. These, however, restrict the type of dependencies. **Concluding**, this paper shows how causal discovery in autocorrelated time series benefits from increasing the effect size of CI tests by including causal parents in conditioning sets. The LPCMCI algorithm introduced here implements this idea by entangling the removal and orientation of edges. As demonstrated in extensive simulation studies, LPCMCI achieves much higher recall than the SVAR-FCI and SVAR-RFCI baselines for autocorrelated continuous variables. We further presented novel orientation rules and an extension of graphical terminology by the notions of middle marks and weakly minimal separating sets. Code for all studied methods is provided as part of the *tigramite* Python package at https://github.com/jakobrunge/tigramite. **In future work** one may relax assumptions of LPCMCI to allow for selection bias and non-stationarity. Background knowledge about (non-)ancestorships may be included without any conceptual modification. Since the presented orientation rules are applicable at any point and thus able to determine (non-)ancestorships already after having performed only few CI tests, the rules may also be useful for the narrower task of causal feature selection in the presence of hidden confounders. Lastly, it would be interesting to combine the ideas presented here with ideas from the structural causal model framework.

## Broader Impact

Observational causal discovery is especially important for the analysis of systems where experimental manipulation is impossible due to ethical reasons, e.g., in climate research or neuroscience. Our work focuses on the challenging time series case that is of particular relevance in these fields. Understanding causal climate mechanisms from large observational satellite datasets helps climate researchers in understanding and modeling climate change as a main challenge of humanity. Since all code will be published open-source, our methods can be used by anyone. Causal discovery is a rather fundamental topic and we deem the potential for misuse as low.

## Acknowledgments and Disclosure of Funding

We thank the anonymous referees for considered and helpful comments that helped to improve the paper. Thanks also goes to Christoph Käding for proof-reading.

DKRZ (Deutsches Klimarechenzentrum) provided computational resources (grant no. 1083).

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
