[Supplementary Material]

# Supplementary material: High-recall causal discovery for autocorrelated time series with latent confounders

**Andreas Gerhardus**
German Aerospace Center
Institute of Data Science
07745 Jena, Germany
andreas.gerhardus@dlr.de

**Jakob Runge**
German Aerospace Center
Institute of Data Science
07745 Jena, Germany
jakob.runge@dlr.de

In this supplementary material we present a brief overview of the FCI algorithm and related graphical terminology as well as details, proofs, further simulation studies, and figures for illustrating the application to the real data example that have been omitted from the main text for reasons of space.

We always assume that there are no selection variables. When saying that $\mathcal{S}$ is a separating set of $A$ and $B$ the exclusions $A \notin \mathcal{S}$ and $B \notin \mathcal{S}$ are implicit. The term *subset* without the attribute *proper* refers to both proper subsets and the original set itself, although in formulas we make this explicit by using the symbol $\subseteq$ instead of $\subset$. We switch between using variable names such as $X_{t-\tau}^i$ and $X_t^j$ that make the time structure explicit, and generic names such as $A$ and $B$ that do not make this explicit (using generic names does, however, *not imply* that there is no time structure). The precise configurations of numerical experiments are given in the respective panel label and figure caption.

## S1 Relevant graphical terminology and notation

The structural causal model (SCM) in eq. (1) can be graphically represented by its time series graph (also known as full time graph) $\mathcal{G}$ [Spirtes et al., 2000, Pearl, 2000, Peters et al., 2017]. This graph contains a node for each variable in the SCM (we use the words *node* and *variable* interchangeably in this context) and an edge (link, words again used interchangeably) $X_{t-\tau}^i \rightarrow X_t^j$ if and only if $X_{t-\tau}^i \in pa(X_t^j)$. It can be understood as a directed acyclic graph (DAG) with infinite extension and repeating structure along the time axis. The parents $pa(X_t^j, \mathcal{G}) = pa(X_t^j)$ of $X_t^j$ are the set of nodes $X_{t-\tau}^i$ with $X_{t-\tau}^i \rightarrow X_t^j$ in $\mathcal{G}$, the ancestors $an(X_t^j, \mathcal{G})$ are the set of nodes connected to $X_t^j$ by a directed path in $\mathcal{G}$ together with $X_t^j$ itself (so every node is an ancestor of itself), and the adjacencies $adj(X_t^j, \mathcal{G})$ the set of nodes connected to $X_t^j$ by any edge in $\mathcal{G}$. Parents are a special case of ancestors. We call $X_t^j$ a descendant of $X_{t-\tau}^i$ if $X_{t-\tau}^i$ is an ancestor of $X_t^j$ (this implies that every node is a descendant of itself). A link between $X_{t-\tau}^i$ and $X_t^j$ is *lagged* if $\tau > 0$, *contemporaneous* if $\tau = 0$, for $i = j$ we speak of an *autodependency link*, and for $i \neq j$ of a *cross link*.

In the presence of unobserved variables so called maximal ancestral graphs (MAGs) [Richardson and Spirtes, 2002] provide an appropriate graphical language for representing causal relationships. Since in this paper we assume the absence of selection variables, the relevant MAGs $\mathcal{M}$ contain two types of edges: directed '$\rightarrow$' and bidirected '$\leftrightarrow$'. These edges are interpreted as composite objects constituted by the symbols at their ends (edge marks), which can be an (arrow-)head ('>' or '<') or a tail ('-'). These edge marks carry a causal meaning: Tails convey ancestorships in $\mathcal{G}$, i.e., $X_{t-\tau}^i \rightarrow X_t^j$ in $\mathcal{M}$ asserts that $X_{t-\tau}^i \in an(X_t^j, \mathcal{G})$; heads convey non-ancestorships in $\mathcal{G}$, i.e., $X_{t-\tau}^i \rightarrow X_t^j$ and $X_{t-\tau}^i \leftrightarrow X_t^j$ in $\mathcal{M}$ say that $X_t^j \notin an(X_{t-\tau}^i, \mathcal{G})$. As an immediate consequence of time order there cannot be a link $X_{t-\tau}^i \leftarrow X_t^j$ for $\tau > 0$ (an effect cannot precede its cause). Parents, ancestors and adjacencies are defined in the same way as for DAGs, and the spouses $sp(X_t^j, \mathcal{M})$ of $X_t^j$ are the set of nodes $X_{t-\tau}^i$ with $X_{t-\tau}^i \leftrightarrow X_t^j$ in $\mathcal{M}$. Two variables are connected by an edge in $\mathcal{M}$ if and only if they cannot be d-separated by a subset of observed

variables in $\mathcal{G}$, and d-separation in $\mathcal{G}$ restricted to observed variables is equivalent to m-separation in $\mathcal{M}$ [Pearl, 1988, Verma and Pearl, 1990, Richardson and Spirtes, 2002]. The parents (ancestors, adjacencies, spouses) of a set of variables are defined as the union of parents (ancestors, adjacencies, spouses) of the individual variables. Example: $pa(\{A, B\}, \cdot) = pa(A, \cdot) \cup pa(B, \cdot)$.

The Markov equivalence class of a MAG is the set of all MAGs that yield the exact same set of m-separations [Zhang, 2008]. These are graphically represented by partial ancestral graphs (PAGs), in which the set of allowed edge marks is extended by the circle mark '$\circ$' [Zhang, 2008]. Such a graph is said to be a PAG for MAG $\mathcal{M}$ if $i)$ it has the same nodes and adjacencies as $\mathcal{M}$ and if $ii)$ all its non-circle edge marks are shared by all members in the Markov equivalence class of $\mathcal{M}$. It is further said to be *maximally informative* if for all its circle marks there is some member of the equivalence class in which there is a tail instead and some other member in which there is a head instead. The wildcard symbol '$*$' may stand for all three possible edge marks (head, tail, circle). This is a notational device only, there are no '$*$' marks in PAGs.

## S2    Some background on FCI

The Fast Causal Inference (FCI) algorithm is an algorithm for constraint-based causal discovery in the presence of unobserved variables [Spirtes et al., 1995, Spirtes et al., 2000, Zhang, 2008]. It allows for both latent confounders and selection variables, although in this paper we assume the absence of selection variables. Under the assumptions of faithfulness [Spirtes et al., 2000], acyclicity, and the existence of an underlying SCM the algorithm determines the maximally informative PAG from perfect statistical decisions of conditional independencies in the distribution $P$ generated by the SCM. The algorithm is based on the following fact:

**Proposition S1** (m-separation by subsets of D-Sep sets [Spirtes et al., 2000]). *Let $A$ and $B$ be two nodes such that $A \notin adj(B, \mathcal{M})$ and $B \notin an(A, \mathcal{M})$, then they are m-separated by some subset of D-Sep$(B, A, \mathcal{M})$. Here:*

**Definition S2** (D-Sep sets [Spirtes et al., 2000]). *Node $V \in \mathcal{M}$ is in D-Sep$(B, A, \mathcal{M})$ if and only if $i)$ it is not $B$ and $ii)$ there is a path $p_V$ between $B$ and $V$ such that $iia)$ all nodes on $p_V$ are in $an(\{A, B\}, \mathcal{M})$ and $iib)$ all non end-point nodes on $p_V$ are colliders on $p_V$.*

A node $B$ is a collider on a path $p$ if the two edges on $p$ involving $B$ both have a head at $B$, as e.g. in $A* \rightarrow B \leftarrow *C$, otherwise it is a non-collider. Together with acyclicity Proposition S1 guarantees that non-adjacent variables $A$ and $B$ are m-separated by a subset of D-Sep$(B, A, \mathcal{M})$ or a subset of D-Sep$(A, B, \mathcal{M})$. However, $\mathcal{M}$ is initially unknown and the D-Sep sets cannot be determined without prior work. Therefore, starting from the complete graph over the set of variables, FCI first performs tests of CI given subset of $pa(B, \mathcal{M}')$ and $pa(A, \mathcal{M}')$ where $\mathcal{M}'$ is the (changing) graph that the algorithm operates on. Whenever two variables are found to be conditionally independent given some subset of variables, the edge between them is removed and their separating set is remembered. This removes some, but in general not all false links. Second, the algorithm orients all resulting unshielded triples $A* \!\!-\!\! *B* \!\!-\!\! *C$ in $\mathcal{M}'$ (these are triples $A* \!\!-\!\! *B* \!\!-\!\! *C$ such that $A$ and $C$ are not adjacent) as colliders $A* \rightarrow B \leftarrow *C$ if $B$ is not in the separating set of $A$ and $C$ (rule $\mathcal{R}0$). We note that at this point head marks are not guaranteed to convey non-ancestorships, but those unshielded triples in $\mathcal{M}'$ that are part of $\mathcal{M}$ are oriented correctly. This is enough to determine the Possible-D-Sep sets, see [Spirtes et al., 2000], which are supersets of the D-Sep sets define above. Third, FCI performs tests of CI given subsets of Possible-D-Sep$(B, A, \mathcal{M}')$ and Possible-D-Sep$(A, B, \mathcal{M}')$. This removes all false links. Fourth, all previous orientations are undone, $\mathcal{R}0$ is applied once more and then followed by exhaustive application of the ten rules $\mathcal{R}1$ through $\mathcal{R}10$. Tests of CI are preferentially made given smaller conditioning sets $\mathcal{S}$, i.e., FCI first tests sets with $|\mathcal{S}| = p = 0$, then those with $|\mathcal{S}| = p = 1$ and so on.

## S3    LPCMCI-PAGs

Section 3.2 introduced middle marks and LPCMCI-PAGs. We here give a more formal definition of these notions. Recall that we assume the absence of selection variables.

**Definition S3** (LPCMCI-PAGs). *Consider a simple graph $\mathcal{C}(\mathcal{G})$ over the same set of variables as $\mathcal{M}(\mathcal{G})$ with edges of the type $\overset{*}{\rightarrow}$, $\leftrightarrow$, $\circ \overset{*}{\rightarrow}$, and $\circ \overset{*}{-} \circ$ where the wildcard '$*$' can stand for the five possible middle marks '?', 'L', 'R', '!', or '' (empty). Such $\mathcal{C}(\mathcal{G})$ is a LPCMCI-PAG for $\mathcal{G}$ with respect*

*to total order $<$ if for any probability distribution $P$ that is Markov relative and faithful to $\mathcal{G}$ the following seven conditions hold:*

1. *If $A *\!\!\rightarrow B$ in $\mathcal{C}(\mathcal{G})$, then $B \notin an(A, \mathcal{G})$.*

2. *If $A \stackrel{*}{\rightarrow} B$ in $\mathcal{C}(\mathcal{G})$, then $A \in an(B, \mathcal{G})$.*

3. *If $A \notin adj(B, \mathcal{C}(\mathcal{G}))$, then $A \notin adj(B, \mathcal{M}(\mathcal{G}))$.*

4. *If $A *\!\!\stackrel{L}{-}\!\!* B$ in $\mathcal{C}(\mathcal{G})$ for $A < B$, then $B \notin an(A, \mathcal{G})$ or there is no $\mathcal{S} \subseteq pa(A, \mathcal{M}(\mathcal{G}))$ that m-separates $A$ and $B$ in $\mathcal{M}(\mathcal{G})$.*

5. *If $A *\!\!\stackrel{R}{-}\!\!* B$ in $\mathcal{C}(\mathcal{G})$ for $A < B$, then $A \notin an(B, \mathcal{G})$ or there is no $\mathcal{S} \subseteq pa(B, \mathcal{M}(\mathcal{G}))$ that m-separates $A$ and $B$ in $\mathcal{M}(\mathcal{G})$.*

6. *If $A *\!\!\stackrel{\perp}{-}\!\!* B$ in $\mathcal{C}(\mathcal{G})$, then both $A *\!\!\stackrel{L}{-}\!\!* B$ and $A *\!\!\stackrel{R}{-}\!\!* B$ would be correct.*

7. *If $A *\!\!-\!\!* B$ in $\mathcal{C}(\mathcal{G})$, then $B \in adj(A, \mathcal{M}(\mathcal{G}))$.*

The first two points give the same causal meaning to head and tail edge marks as they have in MAGs and PAGs. We repeat that while this definition involves a fixed total order $<$, its choice is arbitrary and without influence on the conveyed causal information. Moreover, the definition does not depend on time order. Also note that if all middle marks in $\mathcal{C}(\mathcal{G})$ are empty, then $\mathcal{C}(\mathcal{G})$ is a PAG for $\mathcal{M}(\mathcal{G})$ (guaranteed by the first, second, third, and seventh point). Parents, ancestors, descendants, spouses, and adjacencies in $\mathcal{C}(\mathcal{G})$ are defined (and denoted) in the same way as for MAGs and PAGs, i.e., without being influenced by middle marks.

## S4   Orientation rules for LPCMCI-PAGs

The following is a list of rules for orienting edges in LPCMCI-PAGs. These are extensions of the standard FCI rules [Zhang, 2008] as well as the unshielded triple rule and discriminating path rule of RFCI [Colombo et al., 2012]. If a rule proposes to orient the same edge mark as both tail and head, this is resolved by putting a conflict mark 'x' instead. The edge mark wildcard '$*$' is redefined to stand for the circle, head, tail or conflict mark; the second wildcard symbol '$\star$' excludes the conflict mark. For two reasons we explicitly present and prove also those rules that generalize without much modification: To demonstrate their validity for LPCMCI-PAGs, and to show in which cases the rules also apply to structures with conflict marks.

If $X *\!\!\stackrel{*}{-}\!\!* Y *\!\!\stackrel{*}{-}\!\!* Z$ is an unshielded triple we write $\mathcal{S}_{XZ}$ for the separating set of $X$ and $Z$. Many rules require that $\mathcal{S}_{XZ}$ be weakly minimal and $Y \in \mathcal{S}_{XZ}$. In all these case the requirement of weak minimality can be dropped if $X *\!\!-\!\!* Y *\!\!-\!\!* Z$, i.e., if both middle marks on $X *\!\!\stackrel{*}{-}\!\!* Y *\!\!\stackrel{*}{-}\!\!* Z$ are empty. For this reason the standard FCI orientation rules are implied as special cases.

$\underline{\mathcal{R}0'\mathbf{a}}$: For all unshielded triples $A *\!\!\stackrel{*}{-}\!\!* B *\!\!\stackrel{*}{-}\!\!* C$: If $ia$) $A *\!\!-\!\!* B$ or $ib$) $A$ and $B$ are conditionally dependent given $[\mathcal{S}_{AC} \cup pa(\{A, B\}, \mathcal{C}(\mathcal{G}))] \setminus \{A, B,$ nodes in the future of both $A$ and $B\}$, $iia$) $C *\!\!-\!\!* B$ or $iib$) $C$ and $B$ are conditionally dependent given $[\mathcal{S}_{AC} \cup pa(\{C, B\}, \mathcal{C}(\mathcal{G}))] \setminus \{C, B,$ nodes in the future of both $C$ and $B\}$, $iii$) none of the edge mark '$*$'s at $B$ on $A *\!\!\stackrel{*}{-}\!\!* B *\!\!\stackrel{*}{-}\!\!* C$ is '-' or 'x', and $iv$) $B \notin \mathcal{S}_{AC}$, then mark the unshielded triple for orientation as collider $A *\!\!\rightarrow B \leftarrow\!\!* C$. Condition $ib$) need only be checked if not $ia$), $iib$) need only be checked if not $iia$), and $iv$) need only be checked if all previous conditions are true. If $ib$) or $iib$) find a conditional independence, mark the corresponding edge(s) for removal.

$\underline{\mathcal{R}0'\mathbf{b}}$: For all unshielded triples $A *\!\!\rightarrow B \circ\!\!\stackrel{\perp}{-}\!\!\star C$ and for all unshielded triples $A *\!\!\rightarrow B \circ\!\!\stackrel{R}{-}\!\!\star C$ with $B < C$ and for all unshielded triples $A *\!\!\rightarrow B \circ\!\!\stackrel{L}{-}\!\!\star C$ with $B > C$: If $ia$) $A *\!\!\rightarrow B$ or $ib$) $A$ and $B$ are conditionally dependent given $[\mathcal{S}_{AC} \cup pa(\{A, B\}, \mathcal{C}(\mathcal{G}))] \setminus \{A, B,$ nodes in the future of both $A$ and $B\}$, and $ii$) $B \notin \mathcal{S}_{AC}$, then mark the edge between $B$ and $C$ for orientation as $B \leftarrow\!\!* \star C$ (the middle mark remains as it was before). Condition $ib$) need only be checked if not $ia$). If $ib$) finds a conditional independence, mark the corresponding edge for removal.

$\underline{\mathcal{R}0'\mathbf{c}}$: For all unshielded triples $A *\!\!-\!\!* B \circ\!\!\stackrel{\perp}{-}\!\!\star C$ and for all unshielded triples $A *\!\!-\!\!* B \circ\!\!\stackrel{R}{-}\!\!\star C$ with $B < C$ and for all unshielded triples $A *\!\!-\!\!* B \circ\!\!\stackrel{L}{-}\!\!\star C$ with $B > C$: If $B \notin \mathcal{S}_{AC}$, then mark the edge between $B$ and $C$ for orientation as $B \leftarrow\!\!* \star C$ (the middle mark remains as it was before).

$\underline{\mathcal{R}0'\mathbf{d}}$: For all unshielded triples $A\ast\!\!-\!\!\circ B\circ\!\!-\!\!\ast C$ and for all unshielded triples $A\ast\!\!\to B\circ\!\!-\!\!\ast C$: If $B\notin\mathcal{S}_{AC}$, then mark the unshielded triple for orientation as collider $A\ast\!\!\to B\leftarrow\!\!\ast C$.

$\underline{\mathcal{R}1'}$: For all unshielded triples $A\ast\!\!\overset{\ast}{\to}B\circ\!\!\overset{\ast}{-\!\!\!\star}C$: If $\mathcal{S}_{AC}$ is weakly minimal and $B\in\mathcal{S}_{AC}$, then mark the edge between $B$ and $C$ for orientation as $B\overset{\ast}{\to}C$.

$\underline{\mathcal{R}2'}$: For all $A\overset{\ast}{\to}B\ast\!\!\overset{\ast}{\to}C$ with $A\star\!\!\overset{\ast}{-}\circ C$ and for all $A\ast\!\!\overset{\ast}{\to}B\overset{\ast}{\to}C$ with $A\star\!\!\overset{\ast}{-}\circ C$: Mark the edge between $A$ and $C$ for orientation as $A\star\!\!\overset{\ast}{\to}C$.

$\underline{\mathcal{R}3'}$: For all unshielded triples $A\ast\!\!\overset{\ast}{\to}B\overset{\ast}{\leftarrow}\!\!\ast C$ with $A\star\!\!\overset{\ast}{-}\circ D\circ\!\!\overset{\ast}{-\!\!\!\star}C$ and $D\star\!\!\overset{\ast}{-}\circ B$: If $\mathcal{S}_{AC}$ is weakly minimal and $D\in\mathcal{S}_{AC}$, then mark the edge between $D$ and $B$ for orientation as $D\star\!\!\overset{\ast}{\to}B$.

$\underline{\mathcal{R}4'}$: Use the discriminating path rule of [Colombo et al., 2012] with the following modification: When the rule instructs to test whether any pair $(A, B)$ of variables is conditionally independent given any set $\mathcal{S}$, then $i$) if $A$ and $B$ are connected by an edge with empty middle mark do not make this test, and $ii$) else replace $\mathcal{S}$ with $[\mathcal{S}\cup pa(\{A, B\}, \mathcal{C}(\mathcal{G}))]\setminus\{A, B, \text{nodes in the future of both } A \text{ and } B\}$.

$\underline{\mathcal{R}8'}$: For all $A\overset{\ast}{\to}B\overset{\ast}{\to}C$ with $A\circ\!\!\overset{\ast}{-\!\!\!\star}C$: Mark the edge between $A$ and $C$ for orientation as $A\overset{\ast}{\to}C$.

$\underline{\mathcal{R}9'}$: For all $A_1\circ\!\!\ast\!\!\to A_n$ for which $a$) there is an uncovered potentially directed path from $A_1$ to $A_n$ through $A_2, \ldots, A_{n-1}$ (in this order) such that $b$) $A_2$ is not adjacent to $A_n$: If for all $k=1,\ldots,n-1$ $ia$) $A_k\overset{\ast}{\to}A_{k+1}$ or $ib$) $\mathcal{S}_{A_{k+1}A_{k-1}}$ is weakly minimal and $A_k\in\mathcal{S}_{A_{k+1}A_{k-1}}$ (with the convention $A_0 = A_n$), then mark the edge between $A_1$ and $A_n$ for orientation as $A_1\overset{\ast}{\to}A_n$.

$\underline{\mathcal{R}10'}$: For all $A\circ\!\!\ast\!\!\to D$ for which $a$) there is $B_n\overset{\ast}{\to}D\overset{\ast}{\leftarrow}C_m$, $b$) an uncovered potentially directed path $p_B$ from $A\equiv B_0$ to $B_n$ through $B_1,\ldots,B_{n-1}$ (in this order), $c$) an uncovered potentially directed path $p_C$ from $A\equiv C_0$ to $C_m$ through $C_1,\ldots,C_{m-1}$ (in this order) such that $d$) $B_1$ and $C_1$ are not adjacent: If $i$) $\mathcal{S}_{B_1C_1}$ is weakly minimal and $A\in\mathcal{S}_{B_1C_1}$, $ii$) for all $k=0,\ldots,n-2$ $iia$) $B_{k+1}\overset{\ast}{\to}B_{k+2}$ or $iib$) $\mathcal{S}_{B_{k+2}B_k}$ is weakly minimal and $B_{k+1}\in\mathcal{S}_{B_{k+2}B_k}$, and $iii$) for all $k=0,\ldots,m-2$ $iiia$) $C_{k+1}\overset{\ast}{\to}C_{k+2}$ or $iiib$) $\mathcal{S}_{C_{k+2}C_k}$ is weakly minimal and $C_{k+1}\in\mathcal{S}_{C_{k+2}C_k}$, then mark the edge between $A$ and $D$ for orientation as $A\overset{\ast}{\to}D$.

These rules orient edge marks. They are complemented by the following two rules for updating middle marks:

**APR**: (ancestor-parent-rule, see Lemma 1) Replace all edges $A\overset{!}{\to}B$ by $A\to B$, all edges $A\overset{L}{\to}B$ with $A > B$ by $A\to B$, and all edges $A\overset{R}{\to}B$ with $A < B$ by $A\to B$.

**MMR**: (middle-mark-rule) Replace all edges $A\ast\!\!\overset{?}{\to}B$ with $A < B$ by $A\ast\!\!\overset{L}{\to}B$, all edges $A\ast\!\!\overset{?}{\to}B$ with $A > B$ by $A\ast\!\!\overset{R}{\to}B$, all edges $A\ast\!\!\overset{R}{\to}B$ with $A < B$ by $A\ast\!\!\to B$, and all edges $A\ast\!\!\overset{L}{\to}B$ with $A > B$ by $A\ast\!\!\to B$.

## S5  Pseudocode for Algorithms S2 and S3

In Sec. 3.4 of the main text we give pseudocode for LPCMCI in Algorithm 1. This involves calls to Algorithms S2 and S3, for which we here provide pseudocode and further explanations.

**Algorithm S2** removes the edges between all pairs $(X_{t-\tau}^i, X_t^j)$ of variables that are not adjacent in $\mathcal{M}(\mathcal{G})$ and for which one of them is an ancestor of the other (it may also removed edges between some pairs of non-adjacent variables for which neither one of them is ancestor of the other, but this is not guaranteed). To this end the algorithm tests for CI given $\mathcal{S}\cup\mathcal{S}_{def}$, where the cardinality $|\mathcal{S}| = p$ of $\mathcal{S}\subseteq\mathcal{S}_{search} = apds_t(X_t^j, X_{t-\tau}^i, \mathcal{C}(\mathcal{G}))\setminus\mathcal{S}_{def}$ is successively increased. The $apds_t$ sets are defined in Sec. S7 below, they exclude all variables that have already been identified as non-ancestors of $X_t^j$. This reflects the first design principle behind LPCMCI, see Sect. 3.1. The default conditioning set $\mathcal{S}_{def} = pa(\{X_{t-\tau}^i, X_t^j\}, \mathcal{C}(\mathcal{G}))$ consists of all variables that have been marked as parents of $X_{t-\tau}^i$ or $X_t^j$ in $\mathcal{C}(\mathcal{G})$, which implies that they are ancestors of $X_{t-\tau}^i$ or $X_t^j$ in $\mathcal{G}$. The extension of $\mathcal{S}$ to $\mathcal{S}\cup\mathcal{S}_{def}$ reflects the second design principle behind LPCMCI, see Sect. 3.1, and according to Lemma S4 cannot destroy m-separations. The parentships used to define $\mathcal{S}_{def}$ are found by the application of orientation rules in line 18 (with Alg. S4, see further below in this section) that are made if at least one edge was removed in the current step of the repeat-loop (or have been passed on from an earlier iteration in the preliminary phase of LPCMCI). It is then necessary to restart with $p = 0$, otherwise future separating sets might not be weakly minimal. The rules may also find non-ancestorships, these then further restrict the $apds_t$ sets. Another novelty is that some edges are tested and removed (if

---

**Algorithm S2** Ancestral removal phase

---

**Require:** LPCMCI-PAG $\mathcal{C}(\mathcal{G})$, memory of minimal test statistic values $I^{\min}(\cdot, \cdot)$, memory of separating sets SepSet$(\cdot, \cdot)$, time series dataset $\mathbf{X} = \{\mathbf{X}^1, \dots, \mathbf{X}^N\}$, maximal considered time lag $\tau_{\max}$, significance level $\alpha$, CI test CI$(X, Y, \mathcal{S})$

1: **repeat** starting with $p = 0$
2:     **for** $-1 \leq m \leq \tau_{\max}$ **do**
3:         **for all** ordered pairs of variables $(X^i_{t-\tau}, X^j_t)$ adjacent in $\mathcal{C}(\mathcal{G})$ with $X^i_{t-\tau} < X^j_t$ **do**
4:             **if** $(m = -1$ and $i \neq j)$ or $(m \geq 0$ and $\tau \neq m$ or $i = j)$ **then** continue with next pair
5:             $\mathcal{S}_{def} = pa(\{X^i_{t-\tau}, X^j_t\}, \mathcal{C}(\mathcal{G}))$
6:             **if** the middle mark is '?' or 'L' **then**
7:                 $\mathcal{S}_{search} = apds_t(X^j_t, X^i_{t-\tau}, \mathcal{C}(\mathcal{G})) \setminus \mathcal{S}_{def}$, ordered according to $I^{\min}(X^j_t, \cdot)$
8:                 **if** $|\mathcal{S}_{search}| < p$ **then** update middle mark with 'R' according to Lemma S8
9:                 **for all** subsets $\mathcal{S} \subseteq \mathcal{S}_{search}$ with $|\mathcal{S}| = p$ **do**
10:                     $(p\text{-value}, I) \leftarrow \text{CI}(X^i_{t-\tau}, X^j_t, \mathcal{S} \cup \mathcal{S}_{def})$
11:                     $I^{\min}(X^i_{t-\tau}, X^j_t) = I^{\min}(X^j_t, X^i_{t-\tau}) = \min(|I|, I^{\min}(X^i_{t-\tau}, X^j_t))$
12:                     **if** $p$-value $> \alpha$ **then**
13:                         mark edge for removal, add $\mathcal{S} \cup \mathcal{S}_{def}$ to SepSet$(X^i_{t-\tau}, X^j_t)$
14:                         break innermost for-loop
15:             repeat lines 6 - 14 with $X^i_{t-\tau}$ and $X^j_t$ as well as 'R' and 'L' swapped
16:         remove all edges that are marked for removal from $\mathcal{C}(\mathcal{G})$
17:     **if** any edge has been removed in line 16 **then**
18:         run Alg. S4 using [APR, MMR, $\mathcal{R}8'$, $\mathcal{R}2'$, $\mathcal{R}1'$, $\mathcal{R}9'$, $\mathcal{R}10'$], orient lagged links only
19:         let $p = 0$
20:     **else** increase $p$ to $p + 1$
21: **until** there are no other middle marks than '!' or '' (empty)
22: run Alg. S4 using [APR, MMR, $\mathcal{R}8'$, $\mathcal{R}2'$, $\mathcal{R}1'$, $\mathcal{R}0'd$, $\mathcal{R}0'c$, $\mathcal{R}3'$, $\mathcal{R}4$, $\mathcal{R}9'$, $\mathcal{R}10'$, $\mathcal{R}0'b$, $\mathcal{R}0'a$]
23: **return** $\mathcal{C}(\mathcal{G})$, $I^{\min}(\cdot, \cdot)$, SepSet$(\cdot, \cdot)$

---

found insignificant) before other edges are tested, see lines 2, 4 and the indentation of line 16. To be precise: All autodependency links are tested first, followed by cross links starting with lag $\tau = 0$ and moving to lag $\tau = \tau_{\max}$ in steps of one. This ordering does not depend on the ordering of the $N$ time series variables $X^j$ and does therefore not introduce order-dependence in the sense studied in [Colombo and Maathuis, 2014]. The algorithm converges once all middle marks in $\mathcal{C}(\mathcal{G})$ are '!' or empty. By means of the APR rule (see Lemma 1 or Sect. S4) all edges with a tail mark will then have an empty middle mark, i.e., they cannot be m-separated and do not need further testing. Line 11 updates a memory for keeping track of the minimum test statistic value across all previous CI tests for a given pair of variables (the memory is initialized to plus infinity when line 1 of Algorithm 1 is executed). These values are used to sort $\mathcal{S}_{search}$ in line 7 such that $X^l_{t-\tau_l}$ appears before $X^k_{t-\tau_k}$ in $\mathcal{S}_{search}$ if $I^{\min}(X^j_t, X^l_{t-\tau_l}) > I^{\min}(X^j_t, X^k_{t-\tau_k})$. Note that in line 18 only a select subset of rules is applied and that these are only used to orient lagged links. Moreover, in line 22 we choose to apply the standard rule $\mathcal{R}4$ rather than the modified rule $\mathcal{R}4'$. The reason for this is that, as observed in [Colombo and Maathuis, 2014], the discriminating path rule (on which $\mathcal{R}4'$ is based) becomes computationally intensive when applied in an order-independent way involving conflict resolution. We found these choices to work well in practice but do not claim their optimality.

**Algorithm S3** is structurally similar to Algorithm S2. Once called in line 6 of Algorithm 1, all middle marks in $\mathcal{C}(\mathcal{G})$ are '!' or empty. Whereas edges with empty middle mark are in $\mathcal{M}(\mathcal{G})$ for sure, some edges with middle mark '!' might not be in $\mathcal{M}(\mathcal{G})$. Those latter type of edges are between pairs of variables in which neither one of them is ancestor of the other. According to Lemma S3 below in combination with Proposition S1 such pairs are m-separated by some subset of $napds_t(X^j_t, X^i_{t-\tau}, \mathcal{C}(\mathcal{G}))$ as well as by some subset of $napds_t(X^i_{t-\tau}, X^j_t, \mathcal{C}(\mathcal{G}))$. These sets are defined in Sec. S7 below, they are the more restricted LPCMCI equivalent of the Possible-D-Sep sets in FCI and the $pds_t$ sets in SVAR-FCI. For computational reasons the algorithm nevertheless only searches for separating sets in $napds_t(X^j_t, X^i_{t-\tau}, \mathcal{C}(\mathcal{G}))$, unless for $\tau = 0$ where order-independence dictates otherwise. This is the reason for the logical or-connection in line 10. As compared to

---

**Algorithm S3** Non-ancestral removal phase

---

**Require:** LPCMCI-PAG $\mathcal{C}(\mathcal{G})$, memory of minimal test statistic values $I^{\min}(\cdot,\cdot)$, memory of separating sets SepSet$(\cdot,\cdot)$, time series dataset $\mathbf{X} = \{\mathbf{X}^1, \ldots, \mathbf{X}^N\}$, maximal considered time lag $\tau_{\max}$, significance level $\alpha$, CI test CI$(X, Y, \mathcal{S})$

1: **repeat** starting with $p = 0$
2:    **for** $-1 \leq m \leq \tau_{\max}$ **do**
3:       **for all** ordered pairs of variables $(X^i_{t-\tau}, X^j_t)$ adjacent in $\mathcal{C}(\mathcal{G})$ with $X^i_{t-\tau} < X^j_t$ **do**
4:          **if** the middle mark is empty **then** continue with next pair
5:          **if** $(m = -1$ and $i \neq j)$ or $(m \geq 0$ and $\tau \neq m$ or $i = j)$ **then** continue with next pair
6:          $\mathcal{S}^1_{def} = pa(\{X^i_{t-\tau}, X^j_t\}, \mathcal{C}(\mathcal{G}))$
7:          $\mathcal{S}^2_{def} = $ nodes that have ever been in $pa(\{X^i_{t-\tau}, X^j_t\}, \mathcal{C}(\mathcal{G}))$ since re-initialization
8:          $\mathcal{S}^1_{search} = napds_t(X^j_t, X^i_{t-\tau}) \setminus (\mathcal{S}^1_{def} \cup \mathcal{S}^2_{def})$, ordered according to $I^{\min}(X^j_t, \cdot)$
9:          $\mathcal{S}^2_{search} = napds_t(X^i_{t-\tau}, X^j_t) \setminus (\mathcal{S}^1_{def} \cup \mathcal{S}^2_{def})$, ordered according to $I^{\min}(X^i_{t-\tau}, \cdot)$
10:         **if** $|\mathcal{S}^1_{search}| < p$ or $\tau = 0$ and $|\mathcal{S}^2_{search}| < p$ **then**
11:            Update middle mark with '' according to Lemma S8, continue with next pair
12:         **for all** subsets $\mathcal{S} \subseteq \mathcal{S}^1_{search}$ with $|\mathcal{S}| = p$ **do**
13:            $\mathcal{S}_{def} = \mathcal{S}^1_{def} \cup [\mathcal{S}^2_{def} \cap napds_t(X^j_t, X^i_{t-\tau}, \mathcal{C}(\mathcal{G}))]$
14:            ($p$-value, $I$) $\leftarrow$ CI$(X^i_{t-\tau}, X^j_t, \mathcal{S} \cup \mathcal{S}_{def})$
15:            $I^{\min}(X^i_{t-\tau}, X^j_t) = I^{\min}(X^j_t, X^i_{t-\tau}) = \min(|I|, I^{\min}(X^i_{t-\tau}, X^j_t))$
16:            **if** $p$-value $> \alpha$ **then**
17:               mark edge for removal, add $\mathcal{S} \cup \mathcal{S}_{def}$ to SepSet$(X^i_{t-\tau}, X^j_t)$
18:               break innermost for-loop
19:         **if** $\tau = 0$ **then**
20:            run lines 12 - 18 with $\mathcal{S}^2_{search}$ replacing $\mathcal{S}^1_{search}$, and $X^i_{t-\tau}$ and $X^j_t$ swapped
21:       remove all edges that are marked for removal from $\mathcal{C}(\mathcal{G})$
22:    **if** any edge has been removed in line 21 **then**
23:       run Alg. S4 using the same rules as in line 22 of Alg. S2
24:       let $p = 0$
25:    **else** increase $p$ to $p + 1$
26: **until** all middle marks in $\mathcal{C}(\mathcal{G})$ are empty
27: run Alg. S4 using the same rules as in line 22 of Alg. S2
28: **return** $\mathcal{C}(\mathcal{G})$, $I^{\min}(\cdot, \cdot)$, SepSet$(\cdot, \cdot)$

---

Algorithm S2, the default conditioning is extended: According to Definition S3 a tail on an edge in $\mathcal{C}(\mathcal{G})$ signifies ancestorship in $\mathcal{G}$. Since $\mathcal{C}(\mathcal{G})$ is an LPCMCI-PAG at every point of LPCMCI, $X^i_{t-\tau}$ is an ancestor of $X^j_t$ if there ever was the link $X^i_{t-\tau} \ast\!\!\to X^j_t$. This gives rise to the set $\mathcal{S}^2_{def}$ in line 6. In addition to the parents in $\mathcal{C}(\mathcal{G})$, the algorithm also conditions per default on all nodes in $\mathcal{S}^2_{def}$ that are in the current $napds_t$ set. This decreases the number of sets $\mathcal{S}$ that need to be searched through in the for-loop in line 12 at the price of a higher-dimensional conditioning set. Also this extended default conditioning cannot destroy m-separations. Non-ancestorships are used to constrain the $napds_t$ sets in the first place, and prior to determining $napds_t$ sets the collider rule $\mathcal{R}0'a$ must have been applied to all unshielded triples in $\mathcal{C}(\mathcal{G})$. The algorithm converges once all middle marks are empty, followed by a final exhaustive rule application to guarantee completeness.

**Algorithm S4** exhaustively applies a given set of orientation rules specified by an ordered list $\mathfrak{r}$. The rules are executed in this order and, once any rule has modified $\mathcal{C}(\mathcal{G})$, the loop jumps back to the first rule. This can be used for a preferential execution of simpler and less time consuming rules. Rules $\mathcal{R}0'a$ and $\mathcal{R}0'b$ involve CI tests and may therefore remove some edges. The corresponding separating sets are not guaranteed to be weakly minimal, see Example 1 in the supplement paper to [Colombo et al., 2012] for a counterexample. (There this example is used to show that the separating sets may not be minimal, however it is also a counterexample for weak minimality.) Since many other rules require weak minimality of separating sets, line 10 instructs to make them weakly minimal. This is implemented in the following way: A separating set of $X^i_{t-\tau}$ and $X^j_t$ that is not necessarily

---

**Algorithm S4** Orientation phase

---

**Require:** LPCMCI-PAG $\mathcal{C}(\mathcal{G})$, ordered list of rules $\mathfrak{r}$, memory of minimal test statistic values $I^{\min}(\cdot, \cdot)$, memory of separating sets SepSet$(\cdot, \cdot)$, time series dataset $\mathbf{X} = \{\mathbf{X}^1, \ldots, \mathbf{X}^N\}$, maximal considered time lag $\tau_{\max}$, significance level $\alpha$, CI test CI$(X, Y, \mathcal{S})$

1: $i = 0$
2: **repeat**
3:     apply the $i$-th rule in $\mathfrak{r}$ to $\mathcal{C}(\mathcal{G})$, do not modify $\mathcal{C}(\mathcal{G})$ yet
4:     **if** the rule proposes any modification **then**
5:         **for all** edges marked for orientation **do**
6:             resolve conflicts among the proposed orientations
7:             apply the conflict resolved orientations $\mathcal{C}(\mathcal{G})$
8:         **for all** edges marked for removal **do**
9:             remove the edge from $\mathcal{C}(\mathcal{G})$
10:            make the corresponding separating set weakly minimal
11:         let $i = 0$
12:     **else** increase $i$ to $i + 1$
13: **until** $i \geq \text{len}(\mathfrak{r})$
14: **return** $\mathcal{C}(\mathcal{G})$, $I^{\min}(\cdot, \cdot)$, SepSet$(\cdot, \cdot)$

---

weakly minimal is made weakly minimal by successively removing single elements that are not known ancestors of $X^i_{t-\tau}$ and $X^j_t$ until the resulting set is no separating set anymore. In particular, there is no need to search through all subsets of the original separating set. The validity of this procedure owes to the equivalence of weak minimality and weak minimality of the second type, see Definition S6 and part 3.) of Lemma S7 below. The algorithm also tests for potential conflicts among the proposed orientations and, if present, resolves them by putting the conflict mark 'x'. Most rules require to know whether certain nodes are or are not in certain separating sets. Queries of the second type (*Is node $B$ not in the separating set of nodes $A$ and $C$?*) are answered by a modified version of the majority rule proposed in [Colombo and Maathuis, 2014]. Our modification consists of $i$) searching for separating sets not in the adjacencies of $A$ and $C$ but rather in the relevant $apds_t$ sets and $ii$) including also those separating sets that were found by Algs. S2 and S3 in the majority vote. The second part of this modification is necessary to guarantee completeness (FCI with the unmodified majority rule is not complete, see Sec. S11 for an example). The modification does not introduce order-independence since $i$) the sets $\mathcal{S}_{search}$, $\mathcal{S}^1_{search}$ and $\mathcal{S}^2_{search}$ are ordered by means of $I^{min}(\cdot, \cdot)$ and since $ii$) line 13 of Alg. S2 and line 17 of Alg. S3 instruct to *add* $\mathcal{S} \cup \mathcal{S}_{def}$ to SepSet$(X^i_{t-\tau}, X^j_t)$ rather than saying *write to*. Point $ii$) is relevant for contemporaneous links: if in the same iteration of Alg. S2 (Alg. S3) a pair of variables is found to be conditionally independent given subsets of both $apds_t(X^i_t, X^j_t, \mathcal{C}(\mathcal{G}))$ and $apds_t(X^i_t, X^j_t, \mathcal{C}(\mathcal{G}))$ (both $napds_t(X^i_t, X^j_t, \mathcal{C}(\mathcal{G}))$ and $napds_t(X^i_t, X^j_t, \mathcal{C}(\mathcal{G}))$), *both* separating sets are remembered. The search for separating sets involves the same default conditioning as in Alg. S2. For queries of the first type (*Is node $B$ in the separating set of nodes $A$ and $C$?*) we distinguish two cases. If $B$ is adjacent to both $A$ and $C$ and the middle mark of both edges is empty, then the query is answered in the same way as queries of the first type. Otherwise, the query is answered solely based on the separating sets found by Algs. S2 and S3. Alternatively, one might also in this second case perform a majority-type search of additional separating sets, albeit restricted to separating sets of minimal cardinality due to the requirement of weak minimality (whereas this restriction is not necessary when $A$ and $B$ as well as $C$ and $B$ are connected by edges with empty middle marks). We do not claim optimality of these choices.

## S6   A variant of LPCMCI without Alg. S3

A variant of LPCMCI can be obtained by skipping the execution of Alg. S3 in line 6. According to Lemma S11 the estimated graph $\mathcal{C}(\mathcal{G})'$ is then still a LPCMCI-PAG. This implies that all causal information as conveyed by the absence of edges, by the presence of edges with their respective middle marks, as well as by heads and tails as detailed in Sec. S3 remains correct. Lemma S11 further says that $i$) all edges in $\mathcal{C}(\mathcal{G})'$ that are of the form $\overset{*}{\rightarrow}$ are also in $\mathcal{P}(\mathcal{G})$ and that $ii$) if $X^i_{t-\tau}$ and $X^j_t$ are adjacent in $\mathcal{C}(\mathcal{G})'$ but not in $\mathcal{P}(\mathcal{G})$ then neither of these variables is an ancestor of the other. This

is analogous to RFCI-PAGs, see Theorem 3.4 in [Colombo et al., 2012]. This variant of LPCMCI therefore compares to standard LPCMCI as (SVAR-)RFCI compares to (SVAR-)FCI.

As a side remark, even if LPCMCI is interrupted at any arbitrary point it still yields a graph with unambiguous and sound causal interpretation. This is implied by the fact that the graph $\mathcal{C}(\mathcal{G})$ remains and LPCMCI-PAG at every step of the algorithm, which is proven in Lemmas S9 and S12.

## S7 Definition and relevance of $apds_t$ and $napds_t$ sets

As explained in Sec. S5, Algorithms S2 and S3 respectively perform tests of CI given subsets of $apds_t$ sets and $napds_t$ sets. These are defined and motivated here.

In words $apds_t(X_t^j, X_{t-\tau}^i, \mathcal{C}(\mathcal{G}))$ is the set of all non-future adjacencies of $X_t^j$ other than $X_{t-\tau}^i$ that have not already been identified as non-ancestors of $X_t^j$, formally:

**Definition S4** ($apds_t$ sets)**.** *The set $apds_t(X_t^j, X_{t-\tau}^i, \mathcal{C}(\mathcal{G}))$ is the set of all $X_{t-\tau'}^k$ other than $X_{t-\tau}^i$ with $\tau' \geq 0$ that are connected to $X_t^j$ by an edge without head at $X_{t-\tau'}^k$.*

All statements in this and the following definition are with respect to the graph $\mathcal{C}(\mathcal{G})$. The definition of $napds_t$ sets is more involved. It uses already identified (non-)ancestorships, time order and some general properties of D-Sep sets to provide a tighter approximate of the latter than the Possible-D-Sep sets of FCI and $pds_t$ sets of SVAR-FCI do. Formally:

**Definition S5** ($napds_t$ sets)**.** *1.) The set $napds_t(X_t^j, X_{t-\tau}^i, \mathcal{C}(\mathcal{G}))$ is the union of $napds_t^1(X_t^j, X_{t-\tau}^i, \mathcal{C}(\mathcal{G}))$ and $napds_t^2(X_t^j, X_{t-\tau}^i, \mathcal{C}(\mathcal{G}))$. 2.) The set $napds_t^1(X_t^j, X_{t-\tau}^i, \mathcal{C}(\mathcal{G}))$ is $apds_t(X_t^j, X_{t-\tau}^i, \mathcal{C}(\mathcal{G}))$ without all variables $X_{t-\tau'}^k$ that are connected to $X_{t-\tau}^i$ by an edge with tail at $X_{t-\tau}^i$. 3.) The set $napds_t^2(X_t^j, X_{t-\tau}^i, \mathcal{C}(\mathcal{G}))$ is the set of all variables $X_{t-\tau'}^k$ that are connected to $X_t^j$ by a path $p$ with the following properties: i) on $p$ there is no tail at any node other than $X_{t-\tau'}^k$, ii) the middle node of every unshielded triple on $p$ is a collider on $p$, iii) $p$ does not contain $X_{t-\tau}^i$, iv) the node $X_{t-\tilde{\tau}}^l$ adjacent to $X_t^j$ is not connected to $X_{t-\tau}^i$ by an edge with head at $X_{t-\tilde{\tau}}^l$, and is not after $X_{t-\tau}^i$, v) all nodes on $p$ other than $X_t^j$ and $X_{t-\tilde{\tau}}^l$ are not connected to $X_t^j$ or $X_{t-\tau}^i$ by an edge with tail at $X_t^j$ or $X_{t-\tau}^i$, are not at the same time connected to both $X_t^j$ and $X_{t-\tau}^i$ by edges with a head at themselves, and are not after both $X_t^j$ and $X_{t-\tau}^i$.*

The use of the $apds_t$ and $napds_t$ sets in Algorithms S2 and S3 is due to the following result:

**Lemma S3** (Relevance of $apds_t$ and $napds_t$ sets)**.** *Let $A$ and $B$ be such that $A \notin adj(B, \mathcal{M}(\mathcal{G}))$. 1.) If $A \in an(B, \mathcal{G})$ then $apds_t(B, A, \mathcal{C}(\mathcal{G})) \supseteq D\text{-}Sep(B, A, \mathcal{M}(\mathcal{G}))$. 2.) If $B \notin an(A, \mathcal{G})$, $A \notin an(B, \mathcal{G})$ and rule $\mathcal{R}0'a$ has been exhaustively applied to $\mathcal{C}(\mathcal{G})$ then $napds_t(B, A, \mathcal{C}(\mathcal{G})) \supseteq D\text{-}Sep(B, A, \mathcal{M}(\mathcal{G}))$.*

This remains true when Definition S5 is strengthened in the following way: Whenever the definition demands that there be no edge between $X_{t-\tau}^i$ (or $X_t^j$) and some node $X_{t-\tau_m}^m$ with head at $X_{t-\tau_m}^m$, add the requirement that there be a potentially directed path from $X_{t-\tau_m}^m$ to $X_{t-\tau}^i$ (or $X_t^j$).

## S8 Proofs

**Theorem 1** (LPCMCI effect size)**.** *Let $A \ast\!\!\rightarrow B$ (with $A = X_{t-\tau}^i$ and $B = X_t^j$) be a link ($\rightarrow$ or $\leftrightarrow$) in $\mathcal{M}(\mathcal{G})$. Consider the default conditions $\mathcal{S}_{def} = pa(\{A, B\}, \mathcal{M}(\mathcal{G})) \setminus \{A, B\}$ and denote $\mathbf{X}^* = \mathbf{X} \setminus \mathcal{S}_{def}$. Let $\mathbf{S} = \arg\min_{\mathcal{S} \subseteq \mathbf{X}^* \setminus \{A,B\}} I(A; B|\mathcal{S} \cup \mathcal{S}_{def})$ be the set of sets that define LPCMCI's effect size. If i) there is $\mathcal{S}^* \in \mathbf{S}$ with $\mathcal{S}^* \subseteq adj(A, \mathcal{M}(\mathcal{G})) \setminus \mathcal{S}_{def}$ or $\mathcal{S}^* \subseteq adj(B, \mathcal{M}(\mathcal{G})) \setminus \mathcal{S}_{def}$ and ii) there is a proper subset $\mathcal{Q} \subset \mathcal{S}_{def}$ such that $\mathcal{I}(A; B; \mathcal{S}_{def} \setminus \mathcal{Q}|\mathcal{S}^* \cup \mathcal{Q}) < 0$, then*

$$\min_{\mathcal{S} \subseteq \mathbf{X}^* \setminus \{A,B\}} I(A; B|\mathcal{S} \cup \mathcal{S}_{def}) > \min_{\tilde{\mathcal{S}} \subseteq \mathbf{X} \setminus \{A,B\}} I(A; B|\tilde{\mathcal{S}}) . \tag{S1}$$

*If the assumptions are not fulfilled, then (trivially) "$\geq$" holds in eq. (S1).*

**Remark.** *Assuming the link between $X_{t-\tau}^i$ and $X_t^j$ to be of the form $X_{t-\tau}^i \ast\!\!\rightarrow X_t^j$ is no restriction. If $X_{t-\tau}^i \leftarrow X_t^j$ then $\tau = 0$ by time order and we can swap the roles of $X_{t-\tau}^i = X_t^i$ and $X_t^j$.*

**Proof of Theorem 1.** We start the proof of eq. (S1) by splitting up the set $\mathbf{X}$ that occurs on its right hand side as follows:

$$\min_{\mathcal{S} \subseteq \mathbf{X}^* \setminus \{A,B\}} I(A;B|\mathcal{S} \cup \mathcal{S}_{def}) > \min_{\tilde{\mathcal{S}} \subseteq \mathbf{X} \setminus \{A,B\}} I(A;B|\tilde{\mathcal{S}}) \tag{S2}$$

$$\Leftrightarrow \min_{\mathcal{S} \subseteq \mathbf{X}^* \setminus \{A,B\}} I(A;B|\mathcal{S} \cup \mathcal{S}_{def}) > \min_{\tilde{\mathcal{S}} \subseteq \mathbf{X}^* \setminus \{A,B\}} \min_{\mathcal{Q} \subseteq \mathcal{S}_{def}} I(A;B|\tilde{\mathcal{S}} \cup \mathcal{Q}) \tag{S3}$$

Note that for $\mathcal{Q} = \mathcal{S}_{def}$ the right hand side equals the left hand side. Therefore, eq. (S3) becomes trivially true when ">" is replaced by "$\geq$", but as it stands with ">" it is equivalent to

$$\min_{\mathcal{S} \subseteq \mathbf{X}^* \setminus \{A,B\}} I(A;B|\mathcal{S} \cup \mathcal{S}_{def}) > \min_{\tilde{\mathcal{S}} \subseteq \mathbf{X}^* \setminus \{A,B\}} \min_{\mathcal{Q} \subset \mathcal{S}_{def}, \, \mathcal{Q} \neq \mathcal{S}_{def}} I(A;B|\tilde{\mathcal{S}} \cup \mathcal{Q}) \,, \tag{S4}$$

where $\mathcal{Q}$ is now restricted to be a *proper* subset of $\mathcal{S}$. Let, as stated in the theorem, $\mathbf{S}$ be the set of sets that make the left hand side minimal. A *sufficient* condition for eq. (S4) is then the existence of $\mathcal{S}^* \in \mathbf{S}$ such that

$$I(A;B|\mathcal{S}^* \cup \mathcal{S}_{def}) > \min_{\mathcal{Q} \subset \mathcal{S}_{def}, \, \mathcal{Q} \neq \mathcal{S}_{def}} I(A;B|\mathcal{S}^* \cup \mathcal{Q}) \,. \tag{S5}$$

This implies eq. (S4) because the left hand side of eq. (S5) equals the left hand side of eq. (S4) by definition of $\mathbf{S}$ and the right hand side of eq. (S5) is greater or equal than the right hand side of eq. (S4) because of the additional minimum operation in eq. (S4). By subtracting the left hand side of this inequality we get

$$\min_{\mathcal{Q} \subset \mathcal{S}_{def}, \, \mathcal{Q} \neq \mathcal{S}_{def}} \left[ I(A;B|\mathcal{S}^* \cup \mathcal{Q}) - I(A;B|\mathcal{S}^* \cup \mathcal{Q} \cup (\mathcal{S}_{def} \setminus \mathcal{Q})) \right] < 0 \,. \tag{S6}$$

A difference of conditional mutual informations as in this equation defines a trivariate (conditional) interaction information $\mathcal{I}$ [Abramson, 1963, Runge, 2015], such that we can rewrite eq. (S6) as

$$\min_{\mathcal{Q} \subset \mathcal{S}_{def}, \, \mathcal{Q} \neq \mathcal{S}_{def}} \mathcal{I}(A;B;\mathcal{S}_{def} \setminus \mathcal{Q}|\mathcal{S}^* \cup \mathcal{Q}) < 0 \,. \tag{S7}$$

Contrary to conditional mutual information, the (conditional) interaction information can also attain negative values. This happens when an additional condition, here $\mathcal{S}_{def} \setminus \mathcal{Q}$, increases the conditional mutual information between $A$ and $B$. The second assumption of the theorem states that there is a proper subset $\mathcal{Q} \subset \mathcal{S}_{def}$ for which $\mathcal{I}(A;B;\mathcal{S}_{def} \setminus \mathcal{Q}|\mathcal{S}^* \cup \mathcal{Q}) < 0$. This implies eq. (S7) and hence the main equation (S1). □

We now state a Corollary of Theorem 1, which details graphical assumptions that lead to an increase in effect size as required by eq. (S7). Fig. S1 illustrates these graphical criteria.

**Corollary S1** (LPCMCI effect size). *Let $A{*}{\to}B$ (with $A = X_{t-\tau}^i$ and $B = X_t^j$) be a link ($\to$ or $\leftrightarrow$) in $\mathcal{M}(\mathcal{G})$. Consider the default conditions $\mathcal{S}_{def} = pa(\{A,B\}, \mathcal{M}(\mathcal{G})) \setminus \{A,B\}$ and denote $\mathbf{X}^* = \mathbf{X} \setminus \mathcal{S}_{def}$. Let $\mathbf{S} = \arg\min_{\mathcal{S} \subseteq \mathbf{X}^* \setminus \{A,B\}} I(A;B|\mathcal{S} \cup \mathcal{S}_{def})$ be the set of sets that define LPCMCI's effect size.*

*1.) Assume the link is of the form $A{\to}B$. If i) $pa^*(B, \mathcal{M}(\mathcal{G})) = \mathcal{S}_{def} \setminus pa(A, \mathcal{M}(\mathcal{G}))$ is non-empty (in words: $B$ has parents other than $A$ that are not at the same time also parents of A), and ii) there is $\mathcal{S}^* \in \mathbf{S}$ with $\mathcal{S}^* \subseteq adj(A, \mathcal{M}(\mathcal{G})) \setminus \mathcal{S}_{def}$ or $\mathcal{S}^* \subseteq adj(B, \mathcal{M}(\mathcal{G})) \setminus \mathcal{S}_{def}$, and iii) $B \notin an(\mathcal{S}^*, \mathcal{M}(\mathcal{G}))$, and iv) there is no path between $A$ and $pa^*(B, \mathcal{M}(\mathcal{G}))$ that is active given $pa(A, \mathcal{M}(\mathcal{G})) \cup \mathcal{S}^*$, and v) faithfulness holds, then*

$$\min_{\mathcal{S} \subseteq \mathbf{X}^* \setminus \{A,B\}} I(A;B|\mathcal{S} \cup \mathcal{S}_{def}) > \min_{\tilde{\mathcal{S}} \subseteq \mathbf{X} \setminus \{A,B\}} I(A;B|\tilde{\mathcal{S}}) \,. \tag{S8}$$

*2.) Assume the links is of the form $A{\leftrightarrow}B$. The same inequality (S8) holds if the same assumptions $i) - v)$ as stated in 1.) hold or if these assumptions hold with the roles of $B$ and $A$ exchanged.*

*3.) If neither the assumptions of 1.) nor of 2.) are fulfilled, then (trivially) "$\geq$" holds in (S8).*

**Proof of Corollary S1.** Note that eq. (S8) and eq. (S1) are the same. All manipulations that have identified eq. (S7) as a sufficient condition for eq. (S1) under the assumptions of Theorem 1 are still valid under the assumptions of the corollary. Therefore, eq. (S7) is what remains to be shown.

Figure S1: Graph illustrating the two general cases of dependencies between $A = X_{t-\tau}^i$ and $B = X_t^j$ for proving Corollary S1, namely (**A**) $A \to B$ and (**B**) $A \leftrightarrow B$. The multiple connections are to be understood between subsets of the respective sets such that the whole graph is still a MAG, i.e., that no (almost) directed cycles occur and that maximality is not violated. We omit the links within each subset. $\mathcal{S}^* \subseteq \mathbf{X} \setminus \{pa(\{A, B\}, \mathcal{M}(\mathcal{G})), A, B\}$ denotes the conditions that make the LPCMCI effect size minimal. In panel (**A**) magenta connections are excluded by the assumptions of Corollary S1, in panel (**B**) at least all magenta or all blue connections are excluded (they may both be excluded). These exclusions are, however, not sufficient to guarantee the assumptions of Corollary S1.

Since the interaction information is symmetric in its arguments before the "|", eq. (S7) can be cast into the equivalent conditions:

$$\min_{\mathcal{Q} \subset \mathcal{S}_{def}, \, \mathcal{Q} \neq \mathcal{S}_{def}} \mathcal{I}(A; B; \mathcal{S}_{def} \setminus \mathcal{Q} | \mathcal{S}^* \cup \mathcal{Q}) < 0 \tag{S9}$$

$$\Leftrightarrow \min_{\mathcal{Q} \subset \mathcal{S}_{def}, \, \mathcal{Q} \neq \mathcal{S}_{def}} [I(A; B | \mathcal{S}^* \cup \mathcal{Q}) - I(A; B | \mathcal{S}^* \cup \mathcal{Q} \cup (\mathcal{S}_{def} \setminus \mathcal{Q}))] < 0 \tag{S10}$$

$$\Leftrightarrow \min_{\mathcal{Q} \subset \mathcal{S}_{def}, \, \mathcal{Q} \neq \mathcal{S}_{def}} [I(A; \mathcal{S}_{def} \setminus \mathcal{Q} | \mathcal{S}^* \cup \mathcal{Q}) - I(A; \mathcal{S}_{def} \setminus \mathcal{Q} | \mathcal{S}^* \cup \mathcal{Q} \cup \{B\})] < 0 \tag{S11}$$

$$\Leftrightarrow \min_{\mathcal{Q} \subset \mathcal{S}_{def}, \, \mathcal{Q} \neq \mathcal{S}_{def}} [I(\mathcal{S}_{def} \setminus \mathcal{Q}; B | \mathcal{S}^* \cup \mathcal{Q}) - I(\mathcal{S}_{def} \setminus \mathcal{Q}; B | \mathcal{S}^* \cup \mathcal{Q} \cup \{A\})] < 0 \,. \tag{S12}$$

First consider the case $A \to B$ in conjunction with eq. (S11). Independent of which $\mathcal{Q}$ minimizes the left hand side of this equation, a sufficient condition for its validity is the existence of a proper subset $\mathcal{Q} \subset \mathcal{S}_{def}$ for which the following two conditions hold:

$$I(A; \mathcal{S}_{def} \setminus \mathcal{Q} | \mathcal{S}^* \cup \mathcal{Q}) = 0 \iff A \perp\!\!\!\perp \mathcal{S}_{def} \setminus \mathcal{Q} | \mathcal{S}^* \cup \mathcal{Q} \,, \tag{S13}$$

$$I(A; \mathcal{S}_{def} \setminus \mathcal{Q} | \mathcal{S}^* \cup \mathcal{Q} \cup \{B\}) > 0 \iff A \not\perp\!\!\!\perp \mathcal{S}_{def} \setminus \mathcal{Q} | \mathcal{S}^* \cup \mathcal{Q} \cup \{B\} \,. \tag{S14}$$

We choose $\mathcal{Q} = pa(A, \mathcal{M}(\mathcal{G}))$ and hence get $\mathcal{S}_{def} \setminus \mathcal{Q} = pa^*(B, \mathcal{M}(\mathcal{G}))$. Since by assumption *i*) $pa^*(B, \mathcal{M}(\mathcal{G})) = \mathcal{S}_{def} \setminus \mathcal{Q}$ is not empty, $\mathcal{Q} = pa(A, \mathcal{M}(\mathcal{G}))$ is indeed a proper subset of $\mathcal{S}_{def}$. Further, eq. (S13) is true by assumption *iv*) and eq. (S14) is true by the assumption of faithfulness together with the fact that the path $A \to B \leftarrow pa^*(B, \mathcal{M}(\mathcal{G}))$ is active given $\mathcal{S}^* \cup pa(A, \mathcal{M}(\mathcal{G})) \cup \{B\}$. Since both conditions in eq. (S13) and eq. (S14) hold for this valid choice of $\mathcal{Q}$, part 1.) of the corollary is proven.

We note that assumption *iii*) is needed: Otherwise conditioning on $\mathcal{S}^*$ opens the path $A \to B \leftarrow pa^*(B, \mathcal{M}(\mathcal{G}))$ since $B$ is an ancestor of a conditioned node, thus assumption *iv*) could not be true. Assumption *iii*) would be violated by the magenta connections shown in Fig. S1.

In the case $A \leftrightarrow B$ we can either utilize eq. (S11) or eq. (S12), depending on whether $B$ or $A$ (or both) contain non-empty non-shared parents for which eq. (S13) and eq. (S14) or the equivalent assumptions with $B$ and $A$ exchanged hold. Lastly, the case $A \leftarrow B$ is covered by part 1.) of this corollary with $B$ and $A$ exchanged. This proves part 2.) of the corollary.

Part 3.) follows because the minimum on the right hand side of eq. (S8) is taken over a superset of the set that the minimum on the left hand side is taken over. $\square$

**Lemma S4** (Inclusion of ancestors in separating sets). *Let $A$ and $B$ be m-separated given $\mathcal{S}$, and let $\mathcal{S}_{def} \subseteq an(\{A, B\}, \mathcal{M}(\mathcal{G})) \setminus \{A, B\}$ be arbitrary. Then, $A$ and $B$ are also m-separated given $\mathcal{S}' = \mathcal{S} \cup \mathcal{S}_{def}$.*

**Proof of Lemma S4.** Assume without loss of generality that $\mathcal{S}_{def}$ is non-empty, else the statement is trivial. First, consider the case $\mathcal{S}_{def} \subseteq an(B, \mathcal{M}(\mathcal{G}))$ and assume $\mathcal{S}'$ did not m-separate $A$ and $B$. This requires the existence of a path $p$ between $A$ and $B$ for which $a1$) at least one non-collider on $p$ is in $\mathcal{S}$ or $a2$) there is a collider on $p$ that is not an ancestor of $\mathcal{S}$, b) none of the non-colliders on $p$ is in $\mathcal{S}'$, and c) all colliders on $p$ are ancestors of $\mathcal{S}'$. Since $\mathcal{S}$ is a proper subset of $\mathcal{S}'$, $a1$) conflicts with b). This means $a2$) must be true, i.e, there is at least one collider on $p$ that is an ancestor of $\mathcal{S}' \setminus \mathcal{S} = \mathcal{S}_{def} \setminus \mathcal{S} \subseteq an(B, \mathcal{M}(\mathcal{G}))$ and hence of $B$. Among all those colliders, let $C$ be the one closest to $A$ on $p$. According to b) the sub-path $p_{AC}$ of $p$ from $A$ to $C$ is then active given $\mathcal{S}$ by construction. Since $C$ is an ancestor of $B$ there is at least one directed path $p_{CB}$ from $C$ to $B$. By definition of $C$ the path $p_{CB}$ does not cross any node in $\mathcal{S}$. Thus, $p_{CB}$ is active given $\mathcal{S}$.

We now construct a path from $A$ to $B$ that is active given $\mathcal{S}$, thereby reaching a contradiction. To this end, let $D$ be the node closest to $A$ on $p_{AC}$ that is also on $p_{CB}$ (such $D$ always exists, because $C$ is on both paths). Consider then the subpath $p_{AD}$ of $p_{AC}$ from $A$ to $D$, and the subpath $p_{DB}$ on $p_{CB}$ from $D$ to $B$. Since $p_{AC}$ and $p_{CB}$ are active given $\mathcal{S}$, also $p_{AD}$ and $p_{DB}$ are active given $\mathcal{S}$. By definition of $D$ the concatenation of $p_{AD}$ and $p_{DB}$ at their common end $D$ gives a path $p_{AB}$ from $A$ to $B$. Since $D$ is a non-collider on $p_{AB}$ (because $p_{DB}$ is out of $D$) and $D$ is not in $\mathcal{S}$ (because else $C$ would be an ancestor of $\mathcal{S}$), $p_{AB}$ is active given $\mathcal{S}$. Contradiction.

Second, since the Lemma does not make any distinction between $A$ and $B$, it is also true in case $\mathcal{S}_{def} \subseteq an(A, \mathcal{M}(\mathcal{G}))$. Third, write $\mathcal{S} = \mathcal{S}_A \dot{\cup} \mathcal{S}_B$ with $\mathcal{S}_A = \mathcal{S} \cap an(A, \mathcal{M}(\mathcal{G}))$ and $\mathcal{S}_B = \mathcal{S} \setminus \mathcal{S}_A \subseteq an(B, \mathcal{M}(\mathcal{G}))$. The statement then follows from applying the already proven special cases twice. $\square$

**Lemma S5** (Exclusion of non-ancestors and future from separating sets). *Let $A$ and $B$ be m-separated given $\mathcal{S}$, and let $\mathcal{U}$ be such that $\mathcal{U} \cap an(\{A, B, \mathcal{S} \setminus \mathcal{U}\}, \mathcal{M}(\mathcal{G})) = \emptyset$. Then, $A$ and $B$ are also m-separated given $\mathcal{S}' = \mathcal{S} \setminus \mathcal{U}$. Two important special cases are: Special case 1.) $\mathcal{U} = \mathcal{S} \setminus an(\{A, B\}, \mathcal{M}(\mathcal{G}))$, which allows to restrict separating sets to ancestors. Special case 2.) $\mathcal{U} = \{all\ nodes\ that\ are\ in\ the\ future\ of\ both\ A\ and\ B\}$, which allows to restrict separating sets to the present and past of the later variable.*

**Proof of Lemma S5.** Assume without loss of generality that $\mathcal{U}$ is non-empty, else the statement is trivial. Assume $\mathcal{S}'$ did not m-separated $A$ and $B$. This requires the existence of a path $p$ between $A$ and $B$ for which $a1$) at least one non-collider on $p$ is in $\mathcal{S}$ or $a2$) there is a collider on $p$ that is not an ancestor of $\mathcal{S}$, b) none of the non-colliders on $p$ is in $\mathcal{S}'$, and c) all colliders on $p$ are ancestors of $\mathcal{S}'$. Since $\mathcal{S}'$ is a proper subset of $\mathcal{S}$, $a2$) conflicts with c). This means $a1$) must be true, i.e., there is a non-collider $D$ on $p$ in $\mathcal{S} \setminus \mathcal{S}' = \mathcal{S} \cap \mathcal{U}$. In particular, $D$ is in $\mathcal{U}$. All nodes on $p$ are ancestors of $A$ or $B$ or of a collider on $p$. If $D$ is an ancestor of a collider on $p$, then by c) it is also an ancestor of $\mathcal{S}' = \mathcal{S} \setminus \mathcal{U}$. This shows that $D$ is also in $an(\{A, B, \mathcal{S} \setminus \mathcal{U}\}, \mathcal{M}(\mathcal{G}))$. Since $\mathcal{U} \cap an(\{A, B, \mathcal{S} \setminus \mathcal{U}\}, \mathcal{M}(\mathcal{G})) = \emptyset$, this is a contradiction.

**Special case 1.)** For $\mathcal{U} = \mathcal{S} \setminus an(\{A, B\}, \mathcal{M}(\mathcal{G}))$ we have $\mathcal{S}' = \mathcal{S} \cap an(\{A, B\}, \mathcal{M}(\mathcal{G}))$ and $an(\{A, B, \mathcal{S} \setminus \mathcal{U}\}, \mathcal{M}(\mathcal{G})) = an(\{A, B\}, \mathcal{M}(\mathcal{G}))$. Hence, the condition is fulfilled. **Special case 2.)** For $\mathcal{U} = \{all\ nodes\ that\ are\ in\ the\ future\ of\ both\ A\ and\ B\}$ we have $an(\{A, B, \mathcal{S} \setminus \mathcal{U}\}, \mathcal{M}(\mathcal{G})) \subseteq \{all\ nodes\ that\ are\ not\ after\ both\ A\ and\ B\}$. Hence, the condition is fulfilled.

Note that if $\mathcal{U}$ fulfills the above condition, a proper subset $\mathcal{U}'$ of $\mathcal{U}$ does not necessarily fulfill the condition as well. Consider the example $A \to C \leftarrow D \leftarrow B$. Here $\mathcal{S} = \{C, D\}$ m-separates $A$ and $B$, and $\mathcal{U} = \mathcal{S}$ fulfills the condition. However, $\mathcal{U}' = \{C\}$ does not. This is why we need to require $\mathcal{U} \cap an(\{A, B, \mathcal{S} \setminus \mathcal{U}\}, \mathcal{M}(\mathcal{G})) = \emptyset$ and not just $\mathcal{U} \cap an(\{A, B\}, \mathcal{M}(\mathcal{G})) = \emptyset$. $\square$

**Lemma S6** (Some properties of D-Sep sets). *Consider two distinct nodes $A, B \in \mathcal{M}(\mathcal{G})$. Let $V \in D\text{-}Sep(B, A, \mathcal{M}(\mathcal{G}))$ and path $p_V$ be as in Definition S2, and denote with $C \neq B$ the node on $p_V$ that is closest to $B$. 1.) If $A \notin adj(B, \mathcal{M}(\mathcal{G}))$, then $p_V$ does not contain $A$. 2.) If $B \notin an(A, \mathcal{G})$ and $p_V$ contains two nodes only, then $C$ is a parent or spouse of $B$. 3.) If $B \notin an(A, \mathcal{G})$ and $p_V$ contains more than two nodes, then $C$ is a spouse of $B$ and ancestor of $A$ 4.) If $A \in an(B, \mathcal{G})$, then $D\text{-}Sep(B, A, \mathcal{M}(\mathcal{G})) = pa(B, \mathcal{M}(\mathcal{G}))$.*

**Proof of Lemma S6. 1.)** Assume $p_V$ did contain $A$. The subpath of $p_V$ from $B$ to $A$ is then an inducing path between $B$ and $A$. Since $A$ and $B$ are not adjacent, this violates maximality of the MAG $M$. **2.)** By construction $V = C$ is adjacent to $B$. Assume $C$ was a child of $B$. Since $C$ must be an ancestor of $A$ or $B$, $C$ must be an ancestor of $A$. Then $B$ is an ancestor of $A$, contrary to the assumption. **3.)** According to the second part $C$ is a parent or spouse of $B$. If $C$ was a parent of $B$, $C$ would be a non-collider on $p_V$. This contradicts the definition of

$p_V$, hence $C$ is a spouse of $B$. Moreover, since $C$ is an ancestor of $A$ or $B$, $C$ is an ancestor of $A$. **4.)** The inclusion D-Sep$(B, A, \mathcal{M}(\mathcal{G})) \supseteq pa(B, \mathcal{M}(\mathcal{G}))$ follows since if $V$ is a parent of $B$ then $V{\rightarrow}B$ is a path $p_V$ as required by the definition. We now show the opposite inclusion D-Sep$(B, A, \mathcal{M}(\mathcal{G})) \subseteq pa(B, \mathcal{M}(\mathcal{G}))$ by showing that $V \in pa(B, \mathcal{M}(\mathcal{G}))$. Case 1: $p_V$ has two nodes only. By the second part of this Lemma $V$ is a parent or spouse of $B$. Assume it was a spouse. Then $V$ must be an ancestor of $A$, which with $A \in an(B, \mathcal{G})$ gives $C \in an(B, \mathcal{G})$. But then $C$ cannot be a spouse of $B$. Case 2: $p_V$ has more than three nodes. By the third part of this Lemma we then get that $C$ is an ancestor of $A$, which agains leads to the contradiction $C \in an(B, \mathcal{G})$. $\qquad\square$

**Proof of Lemma S3. 1.)** $A \in an(B, \mathcal{G})$ gives D-Sep$(B, A, \mathcal{M}(\mathcal{G})) = pa(B, \mathcal{M}(\mathcal{G}))$ by part 4.) of Lemma S6. Consider $C \in pa(B, \mathcal{M}(\mathcal{G}))$. Then, $C$ is adjacent to $B$ in $\mathcal{C}(\mathcal{G})$ with a link that does not have a head at $C$. Moreover, $C$ cannot be after $B$. Since $A$ and $B$ are not adjacent, $C$ cannot be $A$. Hence $C$ in $apds_t(B, A, \mathcal{C}(\mathcal{G}))$. **2.)** Consider $V \in$ D-Sep$(B, A, \mathcal{M}(\mathcal{G}))$ and let the path $p_V$ be as in Definition S2. Case 1: $V$ is a parent of $B$ in $\mathcal{C}(\mathcal{G})$. Then, as the proof of the first part of this Lemma shows, $V \in apds_t(B, A, \mathcal{C}(\mathcal{G}))$. Now assume $V$ was a child of $A$ in $\mathcal{C}(\mathcal{G})$. Then $A \in an(B, \mathcal{G})$, contradicting the assumption. Hence $V \in napds_t^1(B, A, \mathcal{C}(\mathcal{G}))$. Case 2: $V$ is not a parent of $B$ in $\mathcal{C}(\mathcal{G})$. We now show that $p_V$ is a path $p$ as required in 3.) of Definition S5 and hence $V \in napds_t^2(B, A, \mathcal{C}(\mathcal{G}))$. Let $C$ be the node on $p_V$ that is closest to $B$, which by 2.) and 3.) of Lemma S6 is a spouse of $B$. $i)$ is true since all non end-point nodes on $p_V$ are colliders on $p_V$ together with the fact that $C$ is a spouse of $B$. $ii)$ is true for the same reason as $i)$ together with the fact that rule $\mathcal{R}0'a$ has been exhaustively applied, which guarantees that if an unshielded triple is a collider then it will be oriented as a collider. $iii)$ is true by 1.) of Lemma S6. $iv)$ is true since $C$ is an ancestor of $A$ by 3.) of Lemma S6. The second and third part of $v)$ are true since all nodes on $p_V$ are ancestors of $A$ or $B$. For the first part of $v)$ observe that if $V$ is a descendant of $A$ (or $B$) in $\mathcal{C}(\mathcal{G})$, then since $V$ is an ancestor of $A$ or $B$ we would get $A \in an(B, \mathcal{G})$ (or $B \in an(A, \mathcal{G})$), a contradiction. $\qquad\square$

**Definition S6** (Weakly minimal separating sets of the second type). *In MAG $\mathcal{M}(\mathcal{G})$ let $A$ and $B$ be m-separated by $\mathcal{S}$. The set $\mathcal{S}$ is a weakly minimal separating set of $A$ and $B$ of the second type if $i)$ there is a decomposition $\mathcal{S} = \mathcal{S}_1 \dot\cup \mathcal{S}_2$ with $\mathcal{S}_1 \subseteq an(\{A, B\}, \mathcal{M}(\mathcal{G}))$ such that $ii)$ if there is $S \in \mathcal{S}_2$ such that $\mathcal{S}' = \mathcal{S} \setminus S$ is a separating set of $A$ and $B$ then $S \in an(\{A, B\}, \mathcal{M}(\mathcal{G}))$. The pair $(\mathcal{S}_1, \mathcal{S}_2)$ is called a weakly minimal decomposition of $\mathcal{S}$ of the second type.*

**Lemma S7** (Selected properties of weakly minimal separating sets). *1.) $\mathcal{S}$ is a weakly minimal separating set of the second type if and only if its canonical decomposition $(\mathcal{T}_1, \mathcal{T}_2)$ defined by $\mathcal{T}_1 = \mathcal{S} \cap an(\{A, B\}, \mathcal{M}(\mathcal{G}))$ and $\mathcal{T}_2 = \mathcal{S} \setminus \mathcal{T}_1$ is a weakly minimal decomposition of $\mathcal{S}$ of the second type. 2.) If $\mathcal{S}$ is a weakly minimal separating set of $A$ and $B$ of the second type then $\mathcal{S} \subseteq an(\{A, B\}, \mathcal{M}(\mathcal{G})) \subseteq an(\{A, B\}, \mathcal{G})$. 3.) $\mathcal{S}$ is a weakly minimal separating set of the second type if and only if it is a weakly minimal separating set. 4.) $\mathcal{S}$ is a weakly minimal separating set of $A$ and $B$ if and only if it is a separating set of $A$ and $B$ and $\mathcal{S} \subseteq an(\{A, B\}, \mathcal{M}(\mathcal{G})) \subseteq an(\{A, B\}, \mathcal{G})$. 5.) If $\mathcal{S}$ is a non-weakly minimal separating set of $A$ and $B$ then there is a proper subset $\mathcal{S}'$ of $\mathcal{S}$ that is a weakly minimal separating set of $A$ and $B$.*

**Proof of Lemma S7. 1.) if:** The existence of a weakly minimal decomposition of the second type implies weak minimality of the second type. **1.) only if:** By assumption there is some weakly minimal decomposition $(\mathcal{S}_1, \mathcal{S}_2)$ of the second type. By definition of the canonical decomposition and by condition $i)$ in Definition S6 the inclusions $\mathcal{S}_1 \subseteq \mathcal{T}_1$ and hence $\mathcal{S}_2 \supseteq \mathcal{T}_2$ hold. Assume the canonical decomposition were not a weakly minimal decomposition of $\mathcal{S}$ of the second type. Then there is some $S \in \mathcal{T}_2$ such that $\mathcal{S}' = \mathcal{S} \setminus S$ is a separating set. Since $\mathcal{S}_2 \supseteq \mathcal{T}_2$ then also $S \in \mathcal{S}_2$, contradicting the assumption that $(\mathcal{S}_1, \mathcal{S}_2)$ is a weakly minimal decomposition of the second type. **2.)** Since $\mathcal{S}$ is weakly minimal of the second type, its canonical decomposition $(\mathcal{T}_1, \mathcal{T}_2)$ is a weakly minimal decomposition of $\mathcal{S}$ of the second type. We now show that $\mathcal{T}_2$ must be empty. Assume it was not and let $C_1, \ldots, C_n$ be its elements. Since by construction $C_1 \notin an(\{A, B\}, \mathcal{M}(\mathcal{G}))$ and since $(\mathcal{T}_1, \mathcal{T}_2)$ is weakly minimal decomposition of the second type, $A$ and $B$ are not m-separated by $\mathcal{S}' = \mathcal{S} \setminus C_1$. This means there is a path $p$ that is active given $\mathcal{S}'$ and blocked given $\mathcal{S}$. Hence, $C_1$ must be a non-collider on $p$. Together with $C_1 \notin an(\{A, B\}, \mathcal{M}(\mathcal{G}))$ this shows that $C_1$ is ancestor of some collider $D_1$ on $p$, which itself is an ancestor of $\mathcal{S}'$ (else $p$ would not be active given $\mathcal{S}'$). Hence, $C_1$ is an ancestor $\mathcal{S}'$. Since $C_1 \notin an(\{A, B\}, \mathcal{M}(\mathcal{G}))$ and $\mathcal{T}_1 \subseteq an(\{A, B\}, \mathcal{M}(\mathcal{G}))$, $C_1$ is an ancestor of $\{C_2, ..., C_n\}$. If $n = 1$, this is a contradiction already. If $n > 1$ we may without loss of generality assume that $C_1$ is an ancestor of $C_2$. Hence, $C_2$ is not an ancestor of $C_1$. By applying the same argument to $C_2$, we conclude that $C_2$ is an ancestor of $\{C_3, ..., C_n\}$. Repeat this until reaching a contradiction.

This shows $\mathcal{T}_2 = \emptyset$ and hence $\mathcal{S} \subseteq an(\{A, B\}, \mathcal{M}(\mathcal{G})) \subseteq an(\{A, B\}, \mathcal{G})$. **3.) if**: Condition $ii)$ in Definition 1 is clearly stronger than $ii)$ in Definition S6. **3.) only if**: Let $\mathcal{S}$ be a weakly minimal separating set of the second type, for which by part 2.) of this Lemma $\mathcal{S} \subseteq an(\{A, B\}, \mathcal{M}(\mathcal{G}))$. Therefore, $(\mathcal{S}, \emptyset)$ is a weakly minimal decomposition of $\mathcal{S}$, showing that $\mathcal{S}$ is weakly minimal. **4.) if**: $(\mathcal{S}, \emptyset)$ is a weakly minimal decomposition **4.) only if**: This follows from parts 2.) and 3.) of this Lemma. **5.)** According to (the first special case of) Lemma S5 $\mathcal{S}' = \mathcal{S} \cap an(\{A, B\}, \mathcal{M}(\mathcal{G}))$ is a separating set. This $\mathcal{S}'$ is weakly minimal according to part 4.) of this Lemma. $\qquad\square$

**Lemma 1** (Ancestor-parent-rule)**.** *In LPCMCI-PAG $\mathcal{C}(\mathcal{G})$ one may replace* **1.)** *$A \overset{\perp}{\to} B$ by $A \to B$,* **2.)** *$A \overset{L}{\to} B$ for $A > B$ by $A \to B$, and* **3.)** *$A \overset{R}{\to} B$ for $A < B$ by $A \to B$.*

**Proof of Lemma 1. 2.)** By the fourth point in Definition S3, $A \notin an(B, \mathcal{G})$ or there is no $\mathcal{S} \subseteq pa(B, \mathcal{M}(\mathcal{G}))$ that m-separates $A$ and $B$ in $\mathcal{M}(\mathcal{G})$. The first option contradicts $A \overset{L}{\to} B$, so the second option must be true. Since $A \in an(B, \mathcal{G})$ gives D-Sep$(B, A, \mathcal{M}(\mathcal{G})) = pa(B, \mathcal{M}(\mathcal{G}))$ according to part 4.) of Lemma S6, Proposition S1 then implies that $A$ and $B$ are not m-separated by any set. **3.)** Equivalent proof. **1.)** Recall that if $A \overset{\perp}{*} B$ in $\mathcal{C}(\mathcal{G})$, then both $A \overset{L}{*} B$ and $A \overset{R}{*} B$ would be correct. The statement then follows since either 2.) or 3.) of this Lemma applies. $\qquad\square$

**Lemma 2** (Strong unshielded triple rule)**.** *Let $A*\!\!\overset{*}{*}\!\!*B*\!\!\overset{*}{*}\!\!*C$ be an unshielded triple in LPCMCI-PAG $\mathcal{C}(\mathcal{G})$ and $\mathcal{S}_{AC}$ the separating set of $A$ and $C$.* **1.)** *If i) $B \in \mathcal{S}_{AC}$ and ii) $\mathcal{S}_{AC}$ is weakly minimal, then $B \in an(\{A, C\}, \mathcal{G})$.* **2.)** *Let $\mathcal{T}_{AB} \subseteq an(\{A, B\}, \mathcal{M}(\mathcal{G}))$ and $\mathcal{T}_{CB} \subseteq an(\{C, B\}, \mathcal{M}(\mathcal{G}))$ be arbitrary. If i) $B \notin \mathcal{S}_{AC}$, ii) $A$ and $B$ are not m-separated by $\mathcal{S}_{AC} \cup \mathcal{T}_{AB} \setminus \{A, B\}$, iii) $C$ and $B$ are not m-separated by $\mathcal{S}_{AC} \cup \mathcal{T}_{CB} \setminus \{C, B\}$, then $B \notin an(\{A, C\}, \mathcal{G})$. The conditioning sets in ii) and iii) may be intersected with the past and present of the later variable.*

**Proof of Lemma 2. 1.)** This follows immediately from part 4.) of Lemma S7. **2.)** By the contraposition of Lemma S4 condition $ii)$ implies that $A$ and $B$ are not m-separated by $\mathcal{S}_{AC}$, and similarly $iii)$ implies the same for $C$ and $B$. The additional claims made in the last sentence of the Lemma follow by the contraposition of Lemma S5. The statement then follows from Lemma 3.1 in [Colombo et al., 2012]. Although there minimality of $\mathcal{S}_{AC}$ is stated as an additional assumption, the proof given in the supplement to [Colombo et al., 2012] does not use this assumption. $\qquad\square$

**Proof of the orientation rules given in subsection S4:** Whenever neither a rule consequent nor the hypothetical manipulations involved in its proof require that a certain edge mark be oriented as head or tail, the rule also applies when that edge mark is the conflict mark 'x'. This explains the use of '$*$' vs. '$\star$' marks in the rule antecedents. We repeat that if $X*\!\!\overset{*}{*}\!\!*Y*\!\!\overset{*}{*}\!\!*Z$ is an unshielded triple and a rule requires $Y \in \mathcal{S}_{XZ}$ with $\mathcal{S}_{XZ}$ weakly minimal, the requirement of weak minimality may be dropped if $X*\!\!-\!\!*Y*\!\!-\!\!*Z$. This is true since when $X*\!\!-\!\!*Y*\!\!-\!\!*Z$ we can conclude $Y \in an(\{X, Z\}, \mathcal{G})$ from $Y \in \mathcal{S}_{XZ}$ even if $\mathcal{S}_{XZ}$ is not weakly minimal.

$\underline{\mathcal{R}0'\mathbf{a}}$: This follows from the second part Lemma 2. Requirement $iii)$ is irrelevant in the case of perfect statistical decisions, it will then never be true given that $ia)$ or $ib)$ and $iia)$ or $iib)$ are true.

$\underline{\mathcal{R}0'\mathbf{b}}$: Assume $B \in an(C, \mathcal{G})$ were true. By Lemma 1 then $B \to C$ in $\mathcal{C}(\mathcal{G})$ and hence in $\mathcal{M}(\mathcal{G})$. Since one of $ia)$ or $iia)$ is true by assumption, there is a path $p_{AB}$ from $A$ to $B$ that is active given $[\mathcal{S}_{AC} \cup pa(\{A, B\}, \mathcal{C}(\mathcal{G}))] \setminus \{A, B, \text{nodes in the future of both } A \text{ and } B\}$. Due to Lemmas S4 and S5 and since $B \notin \mathcal{S}_{AC}$, $p_{AB}$ is also active given $\mathcal{S}_{AC}$. Since then every subpath of $p_{AB}$ is active given $\mathcal{S}_{AC}$ and since $\mathcal{S}_{AC}$ is a separating set of $A$ and $C$, $C$ cannot be on $p_{AB}$. When appending the edge $B \to C$ to $p_{AB}$ we hence obtain a path $p_{AC}$. Since $B$ is a non-collider on $p_{AC}$ and $B \notin \mathcal{S}_{AC}$, $p_{AC}$ is active given $\mathcal{S}_{AC}$. Contradiction. Hence $B \notin an(C, \mathcal{G})$.

$\underline{\mathcal{R}0'\mathbf{c}}$: Assume $B \in an(C, \mathcal{G})$ were true. By Lemma 1 then $B \to C$ in $\mathcal{C}(\mathcal{G})$ and hence in $\mathcal{M}(\mathcal{G})$. Moreover $A \leftarrow B$ or $A \leftrightarrow B$ or $A \to B$ in $\mathcal{M}(\mathcal{G})$ by assumption. In either case $A$, $B$ and $C$ form an unshielded triple in $\mathcal{M}(\mathcal{G})$ with its middle node $B$ not being a collider. But then $B \in \mathcal{S}_{AC}$. Contradiction. Hence $B \notin an(C, \mathcal{G})$.

$\underline{\mathcal{R}0'\mathbf{d}}$: Since all involved middle marks are empty this is just the standard FCI rule $\mathcal{R}0$.

$\underline{\mathcal{R}1'}$: From the first part of Lemma 2 we get $B \in an(\{A, C\}, \mathcal{G})$. Due to the head at $B$ on its edge with $A$ we know $B \notin an(A, \mathcal{G})$. Hence $B \in an(C, \mathcal{G})$.

$\underline{\mathcal{R}2'}$: Assume $C \in an(A, \mathcal{G})$ were true. Case 1: $A \overset{*}{\to} B*\!\!\overset{*}{\to}C$. Due to transitivity of ancestorship then also $C \in an(B, \mathcal{G})$. This contradicts the head at $C$ on its edge with $B$. Case 2: $A*\!\!\overset{*}{\to}B \overset{*}{\to} C$. Then $B \in an(A, \mathcal{G})$, contradicting the head at $B$ on its link with $A$. Hence $C \notin an(A, \mathcal{G})$.

$\underline{\mathcal{R}3'}$: Assume $B \in an(D, \mathcal{G})$ were true. By applying the first part of Lemma 2 to the unshielded triple $A\star^*\!\circ D\circ^*\!\star C$ we deduce that $D \in an(\{A, C\}, \mathcal{G})$. Thus $B \in an(\{A, C\}, \mathcal{G})$, contradicting at least one of the heads at $B$ in the triple $A*\!*\!\!\rightarrow B\leftarrow\!*\!*C$. Hence $B \notin an(D, \mathcal{G})$.

$\underline{\mathcal{R}4'}$: This follows from Lemma 3.2 in [Colombo et al., 2012] together with $i$) the contrapositions of Lemmas S4 and S5, and $ii$) that a pair of variables which in $\mathcal{C}(\mathcal{G})$ is connected by an edge with empty middle mark then this pair of variables is also adjacent in $\mathcal{M}(\mathcal{G})$.

$\underline{\mathcal{R}8'}$: Transitivity of ancestorship gives $A \in an(C, \mathcal{G})$, hence also $C \notin an(A, \mathcal{G})$.

$\underline{\mathcal{R}9'}$: Assume $A_1 \notin an(A_n, \mathcal{G})$ were true, such that $A_1 \leftrightarrow A_n$. From $ia$) or from $ib$) for $k = 1$ together with the first part of Lemma 2 applied to the unshielded triple $A_n \equiv A_0 \leftrightarrow A_1 *\!*\! A_2$ we then conclude $A_1 \!\!\rightarrow\!\! A_2$. By successive application of Lemma 2 to the unshielded triple $A_{k-1} \!\!\rightarrow\!\! A_k *\!*\! A_{k+1}$ together with $ia$) or from $ib$) for $k = 2, \ldots, n-1$ (in this order) we further conclude $A_k \!\!\rightarrow\!\! A_{k+1}$. This gives $A_1 \in an(A_n, \mathcal{G})$, a contradiction to the assumption. Hence $A_1 \in an(A_n, \mathcal{G})$.

$\underline{\mathcal{R}10'}$: Application of the first part of Lemma 2 to the unshielded triple $B_1 *\!*\! A *\!*\! C_1$ gives $A \in an(\{B_1, C_1\}, \mathcal{G})$. Say, without loss of generality, $A \in an(B_1, \mathcal{G})$. By successive application of Lemma 2 to the unshielded triple $B_k \!\!\rightarrow\!\! B_{k+1} *\!*\! B_{k+2}$ together with $ia$) or from $ib$) for $k = 0, \ldots, n-2$ (in this order) we further conclude $B_{k+1} \!\!\rightarrow\!\! B_{k+2}$. This shows that $A \in an(D, \mathcal{G})$.

**APR**: These are the replacements specified in Lemma 1, which was already proven above.

**MMR**: This follows immediately from the causal meaning of middle marks 'L', 'R', and '!' given in Definition S3. $\qquad\square$

**Lemma S8** (Symbolic middle mark update). *Middle marks can be updated by the symbolic rules '?' + '*' = '*', '*' + '' = '' and 'L' + 'R' = '!'.*

**Proof of Lemma S8.** The first rule follows since the middle mark '?' does not make any statement, hence it is consistent with all other middle marks. The second rule follows since the statement made by the empty middle mark '' implies the statements made by all other middle marks. The third rule follows from the definition of the middle mark '!'. $\qquad\square$

**Lemma S9** (Algorithm S2). *Assume Algorithm S2 is being passed a LPCMCI-PAG $\mathcal{C}(\mathcal{G})$ as well as the assumptions stated in Theorem 2. **1.)** $\mathcal{C}(\mathcal{G})$ remains a LPCMCI-PAG at any point of the algorithm. **2.)** The algorithm converges.*

**Proof of Lemma S9.** Write $A = X^i_{t-\tau}$ and $B = X^j_t$. **1.)** Given faithfulness and perfect statistical decisions, edges are removed if and only if the corresponding nodes are m-separated by some subset of variables. The for-loop in line 3 considers ordered pairs $(A, B)$ only if $A < B$ with respect to the adopted total order $<$. According to Lemma S4 the default conditioning on parents as described by lines 5 and 10 does not destroy any m-separations. The algorithm therefore updates the edge between $A$ and $B$ with middle mark 'R' only if $A$ and $B$ are not m-separated by any subset of $apds_t(B, A, \mathcal{C}(\mathcal{G}))$. Since $pa(B, \mathcal{M}(\mathcal{G})) \subseteq apds_t(B, A, \mathcal{C}(\mathcal{G}))$ holds, $A$ and $B$ are then not m-separated by any subset of $pa(B, \mathcal{M}(\mathcal{G}))$ and the update is correct. Similarly the update with middle mark 'L' is correct. Note that the algorithm resets $p = 0$ once any edge marks have been updated, i.e., once some default conditioning sets may potentially change. Therefore, all separating sets found by the algorithm are weakly minimal. More formally: The default conditioning set $\mathcal{S}_{def}$ corresponds to $\mathcal{S}_1$ in Definition 1, and $\mathcal{S}$ corresponds to $\mathcal{S}_2$. Whenever $\mathcal{S}_1$ changes, the algorithm restarts with $|\mathcal{S}_2| = p = 0$ and keeps increasing $p$ by steps of one. If the algorithm finds that some pair of variables is conditionally independent given $\mathcal{S}_{def} \cup \mathcal{S}$, this pair of variables is not conditionally independent given $\mathcal{S}_{def} \cup \mathcal{S}'$ for a proper subset $\mathcal{S}'$ of $\mathcal{S}$. This is because CI given $\mathcal{S}_{def} \cup \mathcal{S}'$ was tested before and rejected, if it would not have been rejected the edge would have been removed already. The statement then follows from correctness of the orientation rules, which is already proven. **2.)** If $A$ and $B$ are connected by a link with middle mark '?' or 'L', the algorithm keeps testing for CI given subsets of $apds_t(B, A, \mathcal{C}(\mathcal{G}))$ until the link has been removed or updated with middle mark 'R'. Similarly, if $A$ and $B$ are connected by a link with middle mark '?' or 'R', the algorithm keeps testing for CI given subsets of $apds_t(A, B, \mathcal{C}(\mathcal{G}))$ until the link has been removed or update with middle mark 'L'. There is no orientation rule that turns a middle mark '!' back into '?', 'L', or 'R', and there is no orientation rule that modifies an empty middle mark. With the update rules given in Lemma S8 this shows that all remaining edges will eventually have middle marks '!' or '' (empty). Then, the algorithm converges. $\qquad\square$

**Lemma S10** (An implication of middle mark '!'). *Assume $A *\!\stackrel{!}{\ast}\! B$ in LPCMCI-PAG $\mathcal{C}(\mathcal{G})$ but $A \notin adj(B, \mathcal{M}(\mathcal{G}))$. Then: **1.)** $A \notin an(B, \mathcal{G})$ and $B \notin an(A, \mathcal{G})$. **2.)** Assume further that $\mathcal{R}0'a$ has been exhaustively applied to $\mathcal{C}(\mathcal{G})$. Then, $A$ and $B$ are m-separated by a subset of $napds_t(B, A, \mathcal{C}(\mathcal{G}))$ and by a subset of $napds_t(A, B, \mathcal{C}(\mathcal{G}))$.*

**Proof of Lemma S10.** Without loss of generality we can assume that $A < B$. **1.)** Assume $A \in an(B, \mathcal{G})$ were true. Then $A$ and $B$ would be m-separated by some subset of D-Sep$(B, A, \mathcal{M}(\mathcal{G}))$ for which D-Sep$(B, A, \mathcal{M}(\mathcal{G})) = pa(B, \mathcal{M}(\mathcal{G}))$ by 4.) of Lemma S6. This contradicts $A *\!\stackrel{R}{\ast}\! B$ and hence $A *\!\stackrel{!}{\ast}\! B$. Similarly $B \in an(A, \mathcal{G})$ contradicts $A *\!\stackrel{L}{\ast}\! B$ and hence $A *\!\stackrel{!}{\ast}\! B$. **2.)** This follows from the first part together with Lemma S3. $\qquad\square$

**Lemma S11** (LPCMCI without Algorithm S3). *Consider modifying the LPCMCI algorithm 1 by skipping line 6 of its pseudocode, i.e., by not executing Algorithm S3. Denote the graph returned by this modified algorithm as $\mathcal{C}(\mathcal{G})'$. Then: **1.)** $\mathcal{C}(\mathcal{G})'$ is a LPCMCI-PAG. **2.)** If $A\!\rightarrow\! B$ in $\mathcal{C}(\mathcal{G})'$ then $A \in adj(B, \mathcal{M}(\mathcal{G}))$. **3.)** If $A *\!\stackrel{*}{\ast}\! B$ in $\mathcal{C}(\mathcal{G})'$ and $A \notin adj(B, \mathcal{M}(\mathcal{G}))$ then $A \notin an(B, \mathcal{G})$ and $B \notin an(A, \mathcal{G})$.*

**Proof of Lemma S11. 1.)** According to the MMR orientation rule the initialization of $\mathcal{C}(\mathcal{G})$ in line 1 of Algorithm 1 produces an LPCMCI-PAG $\mathcal{C}(\mathcal{G})$. Since Lemma S9 proves that $\mathcal{C}(\mathcal{G})$ is still an LPCMCI-PAG after line 3, this remains true when some parentships are carried over after the re-initialization in line 4. The statement thus follows from Lemma S9. **2.)** The proof of the second part of Lemma S3 implies that all edges in $\mathcal{C}(\mathcal{G})'$ have middle marks '!' or '' (empty). If $A\!\rightarrow\! B$ in $\mathcal{C}(\mathcal{G})'$, then $A \in adj(B, \mathcal{M}(\mathcal{G}))$ by the seventh point in Definition S3. If $A\!\stackrel{!}{\rightarrow}\! B$ in $\mathcal{C}(\mathcal{G})'$, then $A \in adj(B, \mathcal{M}(\mathcal{G}))$ by the APR orientation rule, see Lemma 1. **3.)** According to the proof of the previous point $A *\!\!-\!\!\ast B$ or $A *\!\stackrel{!}{\ast}\! B$, and since $A \notin adj(B, \mathcal{M}(\mathcal{G}))$ the seventh point in Definition S3 further restricts to $A *\!\stackrel{!}{\ast}\! B$. The statement then follows from the first part of Lemma S10. $\qquad\square$

**Lemma S12** (Algorithm S3). *Assume Algorithm S3 is being passed a LPCMCI-PAG $\mathcal{C}(\mathcal{G})$. **1.)** $\mathcal{C}(\mathcal{G})$ remains a LPCMCI-PAG at any point of the algorithm. **2.)** The algorithm converges.*

**Proof of Lemma S12.** Write $A = X^i_{t-\tau}$ and $B = X^j_t$. **1.)** An edge between $A$ and $B$ is updated with the empty middle mark only if $A$ and $B$ are not m-separated by a subset of $napds_t(B, A, \mathcal{C}(\mathcal{G}))$ or $\tau = 0$ and $A$ and $B$ are not m-separated by a subset of $napds_t(A, B, \mathcal{C}(\mathcal{G}))$. Note that $\mathcal{R}0'a$ is exhaustively applied in line 22 of Alg. S2 as well as in line 23 of Alg. S3. According to Lemma S10 the update is then correct. Apart from this the proof parallels the proof of 1.) of Lemma S9. **1.)** If $A$ and $B$ are connected by a link with middle mark '!', the algorithm keeps testing for CI given subsets of $napds_t(B, A, \mathcal{C}(\mathcal{G}))$ and if $\tau = 0$ also given subsets of $napds_t(A, B, \mathcal{C}(\mathcal{G}))$ until the link has been removed or updated with the empty middle mark. There is no orientation rule that turns a middle mark '!' back into '?', 'L', or 'R', and there is no orientation rule that modifies an empty middle mark. With the update rules given in Lemma S8 this shows that all remaining edges will eventually have empty middle marks. Then, the algorithm converges. $\qquad\square$

**Theorem 2** (LPCMCI is sound and complete). *Assume that there is a process as in eq.* (1) *without causal cycles, which generates a distribution $P$ that is faithful to its time series graph $\mathcal{G}$. Further assume that there are no selection variables, and that we are given perfect statistical decisions about CI of observed variables in $P$. Then LPCMCI is sound and complete, i.e., it returns the PAG $\mathcal{P}(\mathcal{G})$.*

**Proof of Theorem 2. Soundness:** According to the MMR orientation rule the initialization of $\mathcal{C}(\mathcal{G})$ in line 1 of Algorithm 1 produces an LPCMCI-PAG $\mathcal{C}(\mathcal{G})$. Since Lemma S9 proves that $\mathcal{C}(\mathcal{G})$ is still an LPCMCI-PAG after line 3, this remains true when some parentships are carried over after the re-initialization in line 4. Stationarity both with respect to orientations and adjacencies is always enforced by construction. The statement then follows from Lemmas S9 and S12 together with the first, second, third, and seventh point in Definition S3. **Completeness:** Note that after convergence of the while loop in Alg. S3 all middle marks in $\mathcal{C}(\mathcal{G})$ are empty. Since according to Lemma S12 this $\mathcal{C}(\mathcal{G})$ is a LPCMCI-PAG, the skeleton of $\mathcal{C}(\mathcal{G})$ agrees with that of $\mathcal{M}(\mathcal{G})$. Note again that stationarity both with respect to orientations and adjacencies is always enforced by construction, and that rules $R5$ through $R7$ do not apply due to the assumption of no selection variables. Completeness then follows since the orientation applied in line 27 of Algorithm S3 contain the FCI orientation rules $R0$ through $R4$ and $R8$ through $R10$ as special cases. $\qquad\square$

**Theorem 3** (LPCMCI is order-independent). *The output of LPCMCI does not depend on the order of the $N$ time series variables $X^j$ (the $j$-indices may be permuted).*

**Proof of Theorem 3.** Both Algorithms S2 and S3 remove edges only after the for-loop over ordered pairs has been completed. The ordering of ordered pairs imposed by the outer for-loop is order-independent. Note that the sets $\mathcal{S}_{search}$, $\mathcal{S}^1_{search}$ and $\mathcal{S}^2_{search}$ are ordered by means of $I^{min}$. Since this is an order-independent ordering, the break commands do not introduce order-dependence. The application of orientation rules is order-independent by construction of Algorithm S4: Orientations and removals are only applied once the rule has been exhaustively applied, and conflicts are removed by means of the conflict mark 'x'. Lastly, as discussed at the end of Sec. S5, also the decision of whether a node is in a separating sets are made in an order-independent way. □

## S9 Further numerical experiments

On the following pages we present various further numerical experiments for evaluating and comparing the performances of LPCMCI and the SVAR-FCI and SVAR-RFCI baselines. For each setup we show results for significance levels $\alpha = 0.01, 0.05$ and, depending on the setup, for different autocorrelation values $a$, numbers of observed variables $N$, maximum time lag $\tau_{\max}$, fraction of unobserved variables $\lambda$, and sample sizes $T$. We focus the discussion on orientation recall and precision, runtimes, and control of false positives.

**Nonlinear experiments with GPDC CI test:** Results for the nonlinear conditional independence test GPDC [Runge et al., 2019] are shown in Figures S2 ($T = 200$) and S3 ($T = 400$). Each figure depicts the results for $N = 3, 5, 10$ observed variables and $\alpha = 0.01, 0.05$ with varying autocorrelation on the x-axis. In these experiments we employ a variant of the model in eq. (3) that features half linear and half nonlinear functions of the form $f_i(x) = (1 + 5xe^{-x^2/20})x$, chosen because these tend to yield stationary dynamics. Further, a third of the noise distributions are randomly chosen to be Weibull distributions with shape parameter 2 (instead of Normal distributions).

We find that also here LPCMCI has much higher adjacency and orientation recall than the SVAR-FCI and SVAR-RFCI baselines, especially for contemporaneous links. Precision is overall comparable, but lagged precision often higher for LPCMCI. For $N = 3$ we observe partially not controlled false positives for all methods.

**Linear experiments for varying number of variables $N$:** In Figures S4-S6 we depict results for varying numbers of observed variables $N$ and $T = 200, 500, 1000$, $a = 0, 0.5, 0.95, 0.99$, and $\alpha = 0.01, 0.05$. Since the fraction of unobserved variables is kept at $\lambda = 0.3$, an increasing number of observed variables $N$ also corresponds to an increasing number of total variables $\tilde{N}$.

For the case of no autocorrelation LCPCMI has slightly higher recall and slightly lower precision at a higher runtime. For intermediate autocorrelation ($a = 0.5$) the results are similar to those for $a = 0$, but SVAR-FCI's runtime is higher. For $N = 3, T = 200, \alpha = 0.01$ false positives are not controlled, but less so for LPCMCI. For higher autocorrelation LPCMCI has 0.2-0.4 higher contemporaneous recall and also substantially higher lagged recall throughout. In the highly autocorrelated regime we observe inflated false positives for SVAR-FCI and SVAR-RFCI due to ill-calibrated CI tests, similar to the PC algorithm as discussed in [Runge et al., 2019].

**Linear experiments for varying maximum time lag $\tau_{\max}$:** Figures S7-S9 show results for varying maximum time lags $\tau_{\max}$ and $T = 200, 500, 1000$, $a = 0, 0.5, 0.95, 0.99$, and $\alpha = 0.01, 0.05$.

For no autocorrelation all methods have almost constant contemporaneous recall, only lagged recall shows a slight decay. Note that the true PAG changes with $\tau_{\max}$. Contemporaneous precision is also largely constant, while lagged precision decreases for all methods. Runtime increases and sharply rises for LPCMCI with $k = 0$, indicating that the edge removal phase of Algorithm S3 is faster for higher $k$, i.e., after several preliminary phases have been run. SVAR-FCI similarly features exploding runtimes for large $\tau_{\max}$, both intermediate and higher autocorrelations. Again, false positives in SVAR-FCI and SVAR-RFCI are not well controlled for small $\tau_{\max}$ and $\alpha = 0.01$.

**Linear experiments for varying sample size $T$:** In Figures S10-S12 we depict results for varying sample sizes $T$ and $N = 3, 5, 10$, $a = 0, 0.5, 0.95, 0.99$, and $\alpha = 0.01, 0.05$.

As expected, both recall and precision increase with $T$. Also runtime increases, but only slowly, except for LPCMCI with $k = 0$ where it explodes for $N = 10$. The higher the autocorrelation, the

better the increase in recall and precision for contemporaneous links. For $N = 3$ lack of false positive control (less so for LCPCMI) is visible for all sample sizes. For strong autocorrelations there is a minor decrease in the orientation precision of contemporaneous links when increasing the sample size from $T = 500$ to $T = 1000$. Since at the same time there sometimes is a slight increase in the orientation precision of lagged links, we do not see a straightforward explanation of this observation.

**Linear experiments for varying the fraction of unobserved variables $\lambda$:**  In Figures S13-S18 we show results for varying fractions of unobserved variables $\lambda$ and $T = 200, 500, 1000$, $N = 3, 5, 10$, $a = 0, 0.5, 0.95, 0.99$, and $\alpha = 0.01, 0.05$.

For no autocorrelation both recall and precision decay, while runtime is almost constant. For intermediate and strong autocorrelation we observe a strong decay in recall (even stronger for contemporaneous links), and a less stronger decay in precision. Runtime is almost constant.

**Linear experiments for the non-time case:**  The previous experiments already cover non-autocorrelated time series, see the various results for $a = 0$. In these cases LPCMCI shows a performance similar to the baselines. Here, we additionally analyze the truly non-time series case.

The numerical experiments are based on a purely contemporaneous variant of the model in eq. (3), namely

$$V_t^j = \sum_i c_i V_t^i + \eta_t^j \quad \text{for} \quad j \in \{1, \ldots, \tilde{N}\} . \tag{S15}$$

Note that the time index $t$ could equally well be dropped, such that the samples here conform to the *i.i.d.* case. Contrary to the previous setup we randomly choose $L = \lfloor 0.3\tilde{N}(\tilde{N} - 1)/2 \rfloor$ linear links (corresponding to non-zero coefficients $c_i$). This results in a graph with a constant link density of 30%, which was not feasible in the time series case since it would lead to non-stationary dynamics. As before, the fraction of unobserved variables is $\lambda = 0.3$.

In Figure S19 we show results for varying numbers of observed variables $N = 10, 20, 30, 40$, sample sizes $T = 100, 200, 500$, and significance levels $\alpha = 0.01, 0.05$. While in the non-time series case SVAR-FCI and SVAR-RFCI respectively reduce to FCI and RFCI, we keep the naming for consistency.

SVAR-FCI, SVAR-RFCI, and LPCMCI($k = 0$) all perform very similar with a slight advantage for SVAR-RFCI, which also has the lowest runtimes. LPCMCI($k = 4$) shows higher recall but also lower precision and partly inflated false positives. A more detailed analysis of this drop in performance of LPCMCI with higher $k$ is subject to future research. We hypothesize that with additional preliminary iterations LPCMCI determines increasingly many false ancestorships, which are then used as default conditions and thereby prevent some true conditional independencies from being found.

**Linear experiments for models with discrete variables:**  Recall that LPCMCI is designed to increase effect sizes (with the aim to in turn increase the detection power of true causal links) by using known causal parents as default conditions (and by avoiding to condition on known non-ancestors). The higher-dimensional conditioning sets come, however, at the price of an increased estimation dimension. This has a counteracting negative effect on detection power. As supported by the various experiments in the main text and above, for continuous variables loosing $\mathcal{O}(1)$ degrees of freedom by default conditioning is negligible to, e.g., $\mathcal{O}(100)$ sample sizes. In this case the positive effect of increased effect sizes prevails. We now study the models with discrete-valued variables, where the increased cardinality of conditioning sets is expected to have a much stronger effect on the estimation dimensions.

The numerical experiments are based on a linear and discretized variant of the model in eq. (3) of the form

$$V_t^j = g_{n_{bin}} \left( a_j V_{t-1}^j + \sum_i c_i V_{t-\tau_i}^i + \left( b_t^j - \tfrac{n_{bin}}{2} \right) \right) \quad \text{for} \quad j \in \{1, \ldots, \tilde{N}\} , \tag{S16}$$

where $b_t^j \sim \text{Bin}(n_{bin}, 0.5)$ with $n_{bin} \in 2\mathbb{Z}$ are Binomial noises. The function $g_{n_{bin}}$ decomposes as $g_{n_{bin}} = g_{n_{bin}}^{(1)} \circ g^{(2)}$, where $g^{(2)}$ rounds to the nearest integer and $g_{n_{bin}}^{(1)}(x) = x$ for $|x| \leq n_{bin}/2$ and $g_{n_{bin}}^{(1)}(x) = \text{sign}(x) \cdot n_{bin}/2$ else. This implies that each variable can take at most $n_{bin} + 1$ values. The choice of parameters $a_j$ and $c_j$ as well as the choice and fraction of unobserved variables are unaltered, i.e., as explained below eq. (3) in the main text.

Statistical tests of (conditional) independencies are performed with a $G$-test, see e.g. section 10.3.1 of [Neapolitan, 2003]. The details of our implementation are as follows: When testing for CI of $X$ and $Y$ given $Z_1, \ldots, Z_n$, a separate contingency table of $X$ and $Y$ is made for each unique value $(z_1^a, \ldots, z_n^a)$ that is assumed by the conditions. Rows and columns in these contingency tables that consist of zero counts only are removed. For each such table we calculate the degrees of freedom and the test statistic as for an unconditional $G$-test and sum those values up. If the total number of degrees of freedom falls below one, the test judges independence.

Figure S20 shows the results for $N = 4$ observed variables, sample size $T = 2000$, and significance levels $\alpha = 0.01, 0.05$ with varying autocorrelation on the x-axis.

The bottom row depicts the case with $n_{bin} = 4$. SVAR-FCI, SVAR-RFCI, and LPCMCI($k = 0$) perform largely similar in terms of adjacency and orientation recall. LPCMCI($k = 0$) has slightly higher lagged recall but also lower precision. LPCMCI($k = 4$) has much lower recall (both regarding adjacencies and orientation) for contemporaneous links and autodependency links, but higher recall for lagged links. It further shows uncontrolled false positives and lower precision. One reason for this loss in performance can be found in the much higher cardinalities of CI tests in LPCMCI($k = 4$). These seem to not only lead to lower power but also ill-calibrated CI tests. For $n_{bin} = 2$, depicted in the top row of Fig. S20, the overall results are similar.

These results can only be regarded as preliminary since there are different choices regarding the implementation of the $G$-test of conditional independence as well as a number of other ways to design discrete-variable models.

**Comparison of LPCMCI to residualization approaches:** The previous results demonstrate that LPCMCI shows strong gains in recall for autocorrelated continuous variables. One might wonder whether in the autocorrelated case SVAR-FCI benefits from a data preprocessing that is targeted at removing autocorrelation.

We test this idea by employing two residualization approaches: First, by fitting independent AR(1) models to each times series and then running SVAR-FCI on the residuals (SVAR-FCI-prewhite). This approach was also tested in the causally sufficient case with no contemporaneous links in [Runge, 2018] [Appendix B, Section 3]. Second, by using Gaussian process regression as proposed in [Flaxman et al., 2015] instead of the AR(1) models (SVAR-FCI-GPwhite). More precisely, for every time series $X^j$ we fit a Gaussian process model of $X^j$ on the time index variable ($t = 0, \ldots, n$ where $n$ is the sample size) and apply SVAR-FCI on the residuals. As a kernel we used the Radial Basis Function with an added unit variance White kernel. The RBF hyperparameters are optimzed as part of the default Python `scikit-klearn` implementation. These results are compared to standard LPCMCI run without prior residualization.

Figure S21 shows the results for $N = 5$ observed variables, sample sizes $T = 200, 500$, and significance levels $\alpha = 0.01, 0.05$ with varying autocorrelation on the x-axis.

Both SVAR-FCI-prewhite and SVAR-FCI-GPwhite increase adjacency and orientation recall as compared to SVAR-FCI. SVAR-FCI-prewhite yields larger gains than SVAR-FCI-GPwhite, but both are still well below LPCMCI. Both SVAR-FCI-prewhite and SVAR-FCI-GPwhite have lower precision than SVAR-FCI, lead to inflated false positives, and excessively increase runtime. We further note that it is not clear what the ground truth MAG and PAG should be after residualization. This seems to require a substantially different theory.

Figure S2: Results of numerical experiments for LPCMCI compared to SVAR-FCI and SVAR-RFCI (all with GPDC CI test [Runge et al., 2019]) for varying autocorrelation $a$ for $T = 200$. The left (right) column shows results for significance level $\alpha = 0.01$ ($\alpha = 0.05$). The rows depict results for $N = 3, 5, 10$ (top and bottom). All parameters are indicated in the upper right of each panel.

Figure S3: Results of numerical experiments for LPCMCI compared to SVAR-FCI and SVAR-RFCI (all with GPDC CI test [Runge et al., 2019]) for varying autocorrelation $a$ for $T = 400$. The left (right) column shows results for significance level $\alpha = 0.01$ ($\alpha = 0.05$). The rows depict results for $N = 3, 5, 10$ (top and bottom). All parameters are indicated in the upper right of each panel.

Figure S4: Results of numerical experiments for LPCMCI compared to SVAR-FCI and SVAR-RFCI (all with ParCorr CI test) for varying number of variables $N$ for $T = 200$. The left (right) column shows results for significance level $\alpha = 0.01$ ($\alpha = 0.05$). The rows depict results for increasing autocorrelation (top to bottom). All parameters are indicated in the upper right of each panel. Some experiments did not converge within 24hrs and are not shown.

Figure S5: Results of numerical experiments for LPCMCI compared to SVAR-FCI and SVAR-RFCI (all with ParCorr CI test) for varying number of variables $N$ for $T = 500$. The left (right) column shows results for significance level $\alpha = 0.01$ ($\alpha = 0.05$). The rows depict results for increasing autocorrelation (top to bottom). All parameters are indicated in the upper right of each panel. Some experiments did not converge within 24hrs and are not shown.

Figure S6: Results of numerical experiments for LPCMCI compared to SVAR-FCI and SVAR-RFCI (all with ParCorr CI test) for varying number of variables $N$ for $T = 1000$. The left (right) column shows results for significance level $\alpha = 0.01$ ($\alpha = 0.05$). The rows depict results for increasing autocorrelation (top to bottom). All parameters are indicated in the upper right of each panel. Some experiments did not converge within 24hrs and are not shown.

Figure S7: Results of numerical experiments for LPCMCI compared to SVAR-FCI and SVAR-RFCI (all with ParCorr CI test) for varying maximum time lag $\tau_{\max}$ for $T = 200$. The left (right) column shows results for significance level $\alpha = 0.01$ ($\alpha = 0.05$). The rows depict results for increasing autocorrelation (top to bottom). All parameters are indicated in the upper right of each panel.

Figure S8: Results of numerical experiments for LPCMCI compared to SVAR-FCI and SVAR-RFCI (all with ParCorr CI test) for varying maximum time lag $\tau_{\max}$ for $T = 500$. The left (right) column shows results for significance level $\alpha = 0.01$ ($\alpha = 0.05$). The rows depict results for increasing autocorrelation (top to bottom). All parameters are indicated in the upper right of each panel.

Figure S9: Results of numerical experiments for LPCMCI compared to SVAR-FCI and SVAR-RFCI (all with ParCorr CI test) for varying maximum time lag $\tau_{\max}$ for $T = 1000$. The left (right) column shows results for significance level $\alpha = 0.01$ ($\alpha = 0.05$). The rows depict results for increasing autocorrelation (top to bottom). All parameters are indicated in the upper right of each panel.

Figure S10: Results of numerical experiments for LPCMCI compared to SVAR-FCI and SVAR-RFCI (all with ParCorr CI test) for varying sample size $T$ for $N = 3$. The left (right) column shows results for significance level $\alpha = 0.01$ ($\alpha = 0.05$). The rows depict results for increasing autocorrelation (top to bottom). All parameters are indicated in the upper right of each panel.

Figure S11: Results of numerical experiments for LPCMCI compared to SVAR-FCI and SVAR-RFCI (all with ParCorr CI test) for varying sample size $T$ for $N = 5$. The left (right) column shows results for significance level $\alpha = 0.01$ ($\alpha = 0.05$). The rows depict results for increasing autocorrelation (top to bottom). All parameters are indicated in the upper right of each panel.

Figure S12: Results of numerical experiments for LPCMCI compared to SVAR-FCI and SVAR-RFCI (all with ParCorr CI test) for varying sample size $T$ for $N = 10$ . The left (right) column shows results for significance level $\alpha = 0.01$ ($\alpha = 0.05$). The rows depict results for increasing autocorrelation (top to bottom). All parameters are indicated in the upper right of each panel.

Figure S13: Results of numerical experiments for LPCMCI compared to SVAR-FCI and SVAR-RFCI (all with ParCorr CI test) for varying fraction of unobserved variables $\lambda$ for $T = 200$ and $N = 5$. The left (right) column shows results for significance level $\alpha = 0.01$ ($\alpha = 0.05$). The rows depict results for increasing autocorrelation (top to bottom). All parameters are indicated in the upper right of each panel.

Figure S14: Results of numerical experiments for LPCMCI compared to SVAR-FCI and SVAR-RFCI (all with ParCorr CI test) for varying fraction of unobserved variables $\lambda$ for $T = 500$ and $N = 5$. The left (right) column shows results for significance level $\alpha = 0.01$ ($\alpha = 0.05$). The rows depict results for increasing autocorrelation (top to bottom). All parameters are indicated in the upper right of each panel.

Figure S15: Results of numerical experiments for LPCMCI compared to SVAR-FCI and SVAR-RFCI (all with ParCorr CI test) for varying fraction of unobserved variables $\lambda$ for $T = 1000$ and $N = 5$. The left (right) column shows results for significance level $\alpha = 0.01$ ($\alpha = 0.05$). The rows depict results for increasing autocorrelation (top to bottom). All parameters are indicated in the upper right of each panel.

Figure S16: Results of numerical experiments for LPCMCI compared to SVAR-FCI and SVAR-RFCI (all with ParCorr CI test) for varying fraction of unobserved variables $\lambda$ for $T = 200$ and $N = 10$. The left (right) column shows results for significance level $\alpha = 0.01$ ($\alpha = 0.05$). The rows depict results for increasing autocorrelation (top to bottom). All parameters are indicated in the upper right of each panel.

Figure S17: Results of numerical experiments for LPCMCI compared to SVAR-FCI and SVAR-RFCI (all with ParCorr CI test) for varying fraction of unobserved variables $\lambda$ for $T = 500$ and $N = 10$. The left (right) column shows results for significance level $\alpha = 0.01$ ($\alpha = 0.05$). The rows depict results for increasing autocorrelation (top to bottom). All parameters are indicated in the upper right of each panel.

Figure S18: Results of numerical experiments for LPCMCI compared to SVAR-FCI and SVAR-RFCI (all with ParCorr CI test) for varying fraction of unobserved variables $\lambda$ for $T = 1000$ and $N = 10$. The left (right) column shows results for significance level $\alpha = 0.01$ ($\alpha = 0.05$). The rows depict results for increasing autocorrelation (top to bottom). All parameters are indicated in the upper right of each panel.

Figure S19: Results of numerical experiments in the non-time series case (see eq. (S15)) for LPCMCI compared to SVAR-FCI and SVAR-RFCI (all with ParCorr CI test) for varying number of variables $N$. The left (right) column shows results for significance level $\alpha = 0.01$ ($\alpha = 0.05$). The rows depict results for $T = 100, 200, 500$ (top to bottom). All parameters are indicated in the upper right of each panel. Some experiments did not converge within 24hrs and are not shown.

Figure S20: Results of numerical experiments with the discrete-variable model in eq. (S16) for LPCMCI compared to SVAR-FCI and SVAR-RFCI (all with $G$-test of conditional independence) for varying autocorrelation $a$ for $T = 2000$ and $N = 4$ . The left (right) column shows results for significance level $\alpha = 0.01$ ($\alpha = 0.05$). The top (bottom) row depicts the case with $n_{bin} = 2$ ($n_{bin} = 4$) in eq. (S16).

Figure S21: Results of numerical experiments for LPCMCI compared to SVAR-FCI, SVAR-FCI-prewhite, and SVAR-FCI-GPwhite (all with ParCorr CI test) for varying autocorrelation $a$ for $N = 5$. The left (right) column shows results for significance level $\alpha = 0.01$ ($\alpha = 0.05$). The top (bottom) row depicts results for sample size $T = 200$ ($T = 500$). All parameters are indicated in the upper right of each panel.

# S10 Figures illustrating the application to the real data example

This section shows the results that underlie the discussion of the application to the real data example in Sec. 5 of the main text. Figure S22 shows the PAGs estimated by LPCMCI($k$) for $k = 0, \ldots, 3$ and $\alpha = 0.01, 0.05$. The results of LPCMCI($k = 4$), although mentioned in the main text, are not shown because they agree with those of LPCMCI($k = 3$) for both considered values of $\alpha$. Figure S23 shows the PAGs estimated by SVAR-FCI for $\alpha = 0.01, 0.03, 0.05, 0.08, 0.1, 0.3, 0.5, 0.8$. All results are based on ParCorr CI tests and $\tau_{\max} = 2$. The link colors encode the absolute value of the minimal ParCorr test statistic of all CI tests for the respective pair of variables.

Figure S22: PAGs estimated by LPCMCI($k$) on the real data example as described in Sec. 5 of the main text. The respective values of $k$ and $\alpha$ are shown above each individual plot.

Figure S23: PAGs estimated by SVAR-FCI on the real data example as described in Sec. 5 of the main text. The respective values of $\alpha$ are shown above each individual plot.

## S11 Non-completeness of FCI with majority rule

In Sec. S5 it was mentioned that FCI becomes non-complete when its orientation rules in the final orientation phase are modified according to the majority rule of [Colombo and Maathuis, 2014]. While this is probably known, we have not found it spelled out in the literature. Therefore, we here illustrate this point by the example given in Fig. S24.

Figure S24: Example to illustrate the non-completeness of FCI with majority rule. (**A**) MAG $\mathcal{M}$. (**B**) Maximally informative PAG for $\mathcal{M}$, output of FCI *without* majority rule. (**C**) Output of FCI *with* majority rule.

The left and middle part of the figure respectively show the true MAG and its fully informative PAG. As proven in [Zhang, 2008], the latter will be found by the standard FCI algorithm without modification according to the majority rule. Note that the two heads at node $F$ are put by the collider rule $\mathcal{R}0$: Since $F$ is not in the separating set $\mathcal{S}_{DE} = \{A, B, C\}$ of $D$ and $E$, the unshielded triple $D *\!\!-\!\circ F \circ\!\!-\!\!* E$ is oriented as collider $D *\!\!\rightarrow F \leftarrow\!\!* E$. The output of FCI with modification according to the majority rule is shown in the right part of the figure. There, the two heads at $F$ are not found. The reason is that the majority rule instructs $\mathcal{R}0$ to base its decision of whether $D *\!\!-\!\circ F \circ\!\!-\!\!* E$ is oriented as a collider not on the separating set found during the removal phases (this is $\mathcal{S}_{DE}$) but rather on a majority vote of all separating sets of $D$ and $E$ in the adjacencies of $D$ and $E$. However, in the example there are no such separating sets since neither $D$ nor $E$ is adjacent to $A$. Therefore, $D *\!\!-\!\circ F \circ\!\!-\!\!* E$ is not oriented as collider by $\mathcal{R}0$ but rather marked as ambiguous. The heads can also not be found by $\mathcal{R}2$, $\mathcal{R}3$ and $\mathcal{R}4$, the other rules for putting invariant heads, because these only oriented edges that are part of a triangle. Since neither $F \circ\!\!\rightarrow D$ nor $F \circ\!\!\rightarrow E$ is part of a triangle, the orientations are not found. As described at the end of Sec. S5 we employ a modified majority rule in LPCMCI to guarantee both completeness and order-independence.