[Reviews · NeurIPS 2020]

Review 1

Summary and Contributions: The paper provides an RFCI-based algorithm optimized for time-series data, which focuses on conditioning sets in independence tests. It was shown that by including known parents in conditioning sets, and removing non-ancestors, the effect size of independence tests increased, lowering the number of missed true edges in the output graph. This also has the double benefit of decreasing computational complexity by restricting the conditioning sets that need to be checked.

Strengths: The goal of causal discovery is very common when given time-series data, so methods that improve on existing approaches are welcome. While artificial, the experiments did show that the proposed changes to RFCI significantly improve the resulting output graph. An additional asset is the inclusion of Python code implementing the method.

Weaknesses: The paper does not hide the fact that this improvement is, by its nature, iterative - it improves an existing algorithm to work in a wider variety of situations (particularly in autocorrelated timeseries). Nevertheless, due to the prevalence of practical situations in which LPCMCI could improve over existing methods, I think this should not count strongly against the work. However, this is contingent on the algorithm being significantly better than previous work in realistic scenarios. The experiments showed a clear improvement, but were based on artificial data. The difficulty of getting ground-truth data for real situations notwithstanding, it would be a great asset to the paper to compare the algorithms on a real dataset - even if just to verify that the algorithm gives a significant improvement in practical settings.

Correctness: I did not find any errors, but I did not check proofs.

Clarity: The paper is very clear and focused. I appreciated the example demonstrating the limitations of previous methods. I also appreciated the fact that the method's benefits and drawbacks in comparison to previous literature were clearly stated.

Relation to Prior Work: Yes, prior work is described, and is compared to the proposed method in the experimental section. However, I might not be familiar with the full recent literature.

Reproducibility: Yes

Additional Feedback: Having read the author feedback and subsequent discussion, I maintain my belief that the paper is above the acceptance threshold.


Review 2

Summary and Contributions: I've read the authors' feedback. -- Many causal discovery algorithms rely on conditional independence tests of variables, in particular, such algorithms that aim to find Markov equivalence classes without assuming particular functional forms of causal relations. This paper considers to improve the performance of such causal discovery methods in the presence of latent confounders for time series data by improving the performance of conditional independence test. Their key idea is to first condition the parents of variables whose conditional independence is examined rather than every combination of the other variables and extend the conditioning set of parents. This leads to higher recalls of conditional independence tests and hence better performance in estimating causal structures.

Strengths: They aim to improve the performance of a major causal discovery approach based on conditional independence. They consider a general model for time series data allowing latent confounders. They have provided relevant theorems that are necessary when they restrict the default conditioning set in testing the conditional independence of two variables to their parents rather in their basis method FCI (Spirtes et al., 1995). In particular, Theorem 1 tries to explain the reason why restricting the conditioning set to parents could lead to better performance. In their experiments, their method achieved higher recalls of conditional independence tests and this lead to better performance of causal discovery.

Weaknesses: Theorem 1 looks saying that (conditional) mutual information between variables is larger when the conditioning set includes the parents than when it does not. It would be helpful to show how this leads to higher recalls of conditional independence tests.

Correctness: I haven't found serious mistakes.

Clarity: It was not very clear to me how their algorithm first finds parents of variables. I guess that they store such parents when they find from data in their algorithm. Would it be correct?

Relation to Prior Work: It looks ok.

Reproducibility: Yes

Additional Feedback:


Review 3

Summary and Contributions: Thank you to the authors for the detailed feedback and for addressing most of the concerns raised. ==================== The authors tackle the problem of causal discovery from observational autocorrelated stationary time series data in the presence of latent confounders. They argue that classic constraint-based methods for causal discovery, such as the FCI algorithm, are suboptimal when applied to this type of data due to the usual choice of conditional independence (CI) tests. This weakness manifests itself in the form of low recall for the discovery of causal links and the authors show how to mitigate it by choosing a more appropriate set of CI tests to perform. The proposed approach allows for general functional relationships and general types of variables, it can handle both lagged and contemporaneous interactions, and it requires no assumptions to be made regarding the type of unmeasured confounding. The main contributions in this paper are as follows: 1. The authors discuss the importance of effect size (the minimum test statistic) when performing CI tests, as low effect size, and by extension low statistical power, leads to poor recall of graph adjacencies and orientations. They propose a different approach for choosing the conditioning sets that helps to increase the effect size and improve the recall measure. 2. The authors introduce an algorithm (LPCMCI) designed for causal discovery from, but not limited to, stationary multivariate time series data in the presence of latent confounders. The algorithm makes use of new orientation rules that generalize the standard (R)FCI rules and a new class of graphs (LPCMCI-PAG) for representing intermediate stages of orientation. 3. The authors make a theoretical contribution by showing that their new approach is sound, complete, and order-independent.

Strengths: The main strength of this paper is, in my opinion, its novel approach to scheduling conditional independence tests. The authors showcase an interesting problem regarding the effect size of conditioning sets in low signal-to-noise ratio scenarios, due to e.g., autocorrelation. They highlight the fact that the effect size can be increased by choosing more appropriate conditioning sets, often with higher cardinality, which is counterintuitive relative to the more general statistically motivated approach of choosing low-cardinality conditioning sets. Another strength of the paper is the solid grounding in information-theoretic concepts and the theoretical results regarding the soundness, completeness and order-independency of the proposed algorithm. These are accompanied by a comprehensive set of numerical experiments showcasing the expected improvement in recall due to incorporating time series information.

Weaknesses: The work is restricted by the setting assumed. The authors focus on a particular type of data, namely stationary multivariate autocorrelated time series. While the authors show that their approach improves upon the state-of-the-art in this particular setting, it is unclear how well the method works in general. The authors mention the fact that their method is also applicable to non-time series data, although LPCMCI appears specifically designed to work with time series datasets, but do not indicate the gain or loss in performance we should expect compared to traditional approaches in that scenario. I would expect that the more optimal approach would still involve performing conditional independence tests in which the conditioning sets are as small as possible, so I would not expect LPCMCI to work as well as FCI in the general case. Could the authors comment on that aspect? How well do they expect LPCMCI to work for non-time series data? The paper is also lacking evaluation on a real-world dataset. Even though it is generally very difficult to find a good causal discovery application, e.g., in which the ground truth is known, I would have found it useful if the authors had mentioned an example of a dataset (from climate research or neuroscience) on which their proposed approach could successfully be applied.

Correctness: The claims seem reasonable and the methodology appears solid. However, the proofs are almost entirely deferred to the appendix, so the main paper cannot be assessed in isolation.

Clarity: The paper is generally well-written and reasonably well-structured. The authors take their time to highlight important points when necessary, giving the reader much needed breathing space.

Relation to Prior Work: The authors have included a good overview of existing methods for causal discovery, constraint-based or otherwise. They also clearly specify the main shortcomings of each of these methods that they wish to address with their proposed approach. The proposed LPCMCI algorithm is compared against traditional algorithms such as FCI and its variants (RFCI). The authors explain in plenty of detail how the LPCMCI algorithm is designed to handle the constraint-based causal discovery task differently from FCI, with a focus on improving recall when applied to autocorrelated time series data. The two approaches are compared in detail and supported by an extensive set of simulation experiments. The current LPCMCI approach draws heavy inspiration from the PCMCI algorithm (Runge et al., 2019). It is even touted as latent PCMCI, so this naturally begs the question of how this work differs from the previous contribution. I would have liked to see a more explicit comparison between the two approaches. Clearly, PCMCI cannot be applied to the same set of problems described in this paper because it was not designed to handle latent confounders. However, there seem to be fundamental differences between the two approaches that are not sufficiently explained. For instance, an important desideratum for PCMCI seems to be reducing the size of conditioning sets when performing conditional independence tests so as to increase effect size. In LPCMCI on the other hand, it turns out that we can achieve a larger effect size despite increasing the cardinality of conditioning sets. Could the authors elucidate what the key differences are when making the transition from causal sufficiency to causal insufficiency?

Reproducibility: Yes

Additional Feedback: - It might be worthwhile to introduce the term 'auto-link'. - The simulation section is extensive, but perhaps a bit too dense. It would improve readability if the authors could find a way to separate the overlapping legends from the plots.


Review 4

Summary and Contributions: This paper explores learning causal structure of time series where latent confounder might exist. The authors propose to take advantage of a known time-order to orient causal direction as early as possible to conduct conditional independence tests that enjoy higher effect size by conditioning on known parents. This leads to high-recall for adjacencies (especially for contemporaneous connections) among variables and for orientation. The authors demonstrate that the performance gain is more prominent when autocorrelation is stronger.

Strengths: The use of higher effect size conditional independence tests is based on well-established information theory. The empirical performance gain is remarkable. The causal discovery with high recall (and high precision) has always been appreciated by NeurIPS community. In this case, this paper has both theoretical and practical importance.

Weaknesses: The reliability of conditional independence test for checking ‘dependence’ hinges not only on effect size but also cardinality of conditioning set. Further the behavior CI test is quite different for continuous and discrete variables as conditioning variables. Partial correlation is good for a linear case, and there are many kernel-based CI tests (KCIT by Kun Zhang, etc). For discrete variables, the power of CI test will be significantly affected by the size of conditioning set. Although the authors clearly mentioned that (i) CI tests’ estimation dimension (L138) affects the performance, (ii) there exist disadvantages due to increased conditioning (L165–167), and (iii) their use of all known parents as conditioning set is not the best option (L169), the paper does not explore any empirical results based on discrete variable time series despite the fact that authors alluded that their methods also works for discrete case (Line 77). Further some of tests with higher cardinality is avoided by limiting the size of conditioning sets (L276). The use of known parents, here especially the autocorrelation case, seems crucial. But in many time-series based analysis often removes the effect of autocorrelation by, e.g., residualization. (see Flaxman, et al 2016). The use of known parents also appears in Lee and Honavar 2017, see for limiting the separating set to the superset of parents (Section 5 non-RBO, but only for the parents of one variable.) References - Seth R Flaxman, Daniel B. Neill, and Alexander J. Smola. Gaussian processes for independence tests with non- iid data in causal inference. ACM Transactions on Intelligent Systems and Technology, 7(2):1–23, 2016. - Sanghack Lee and Vasant Honavar. Towards Robust Relational Causal Discovery A Kernel Conditional Independence Test for Relational Data. In Proceedings of the Thirty-third Conference on Uncertainty in Artificial Intelligence, 2017.

Correctness: The paper seems to be correct without rigorously examining the correctness of proof in the supplementary material.

Clarity: Due to the amount of required materials (proofs, etc), many parts are written without details. In such case, high-level information still needs to maintain concreteness without being abstract. There are several places where explanations are not clearly understandable without actually knowing the underlying details. Those parts can be improved but the paper is generally written well while conveying important points of the paper clear.

Relation to Prior Work: The previous work on causal discovery for time series is well-summarized and contrasted.

Reproducibility: Yes

Additional Feedback: Thanks for the feedback and perfoming experiments in such a short time period. Many of my concerns are resolved. Yet it is less clear about how good residualization will be (is GP alpha optimized?) and how bad the discrete variables case will be. I am raising my score from 5 to 6. =================== Thanks for a great work. I have a few comments and questions. The paper mentions that the new method will be compared to SVAR-FCI and (time series adaptation of) RFCI. But the paper mentions FCI throughout the rest of the paper. Is “FCI” “SVAR-FCI” or genuine “FCI”? If it is genuine FCI, comparing new method to FCI in the motivation is a little bit absurd to me given that we should focus on “time-series”. For example, consider two time series X and Y. One cannot simply test independence by applying an IID-based CI test on {(X_t, Y_t)} due to its non-iid nature. IID-based test can be applicable when we properly block e.g., X_t-1 and Y_t-1 (assuming that the maximum lag is 1). This is why often we can find the residualizing time series to remove autocorrelation (see above for the weakness). The use of background knowledge to enhance tests can be applied to any data sets not just for a time series (where a time order is available). Whenever there exists background knowledge about parentship, it can be taken into account. However, as you alluded, there are certain disadvantages of conditioning parents of both variables with respect to orientation (or adjacency tests) as already mentioned in L165–L167. If we are worried about false negatives (which will later cause false positives and etc), one can at least do better by not assuming “orientation-faithfulness” (Ramsey et al. 2012). Consider testing for X→Y←Z based on a conditioning set S\{Y}. If the result is negative (independence), one can further test with a conditioning set S\cup {Y}. If it also yields negative, we can avoid prematurely orienting X→Y←Z. A question for experiments. If we look closely at results in supplementary material, there is a small drop of precision from T=500 to T=1000. Do you have any insights to share? Is this because additionally recovered edge with higher recall is relatively weak (since it was not captured with T=500) or is it because of higher cardinality in testing? worsening the precision... minor points. Bold letters seem a bit overused. L140 this are L247 What do you mean by “Stationarity” is enforced? Figure 2 (A)(B)(C)(D) are not labelled in the graph although one can infer. I have no doubt that the paper addresses fundamental issues with previous methods but the extensive empirical section (in the supplementary material) seems missing answering other subtle issues with high-recall CI tests: First, how a larger cardinality in CI tests will affect the performance overall? Second, how tests on the discrete variables affect the results? My current overall score will be adjusted depending on how the authors address issues with larger cardinality, discrete variables, and residualization (or other existing methods for de-autocorrelation)

[Author Response · NeurIPS 2020]

We thank the reviewers for their thorough and inspiring comments. The overall feedback is positive, with the main suggestions for improvement being $i$) an application to real data, $ii$) further tests and discussions (non-time series case, high cardinality, discrete variables), and $iii$) a comparison to residualization approaches. We will follow these suggestions by including further experiments and discussions in the camera-ready version. Below, our brief answers.

**Real data:** We agree that, despite the difficulty of basing the evaluation of causal discovery methods on real data, a real data example would be an asset to the paper. We will therefore include an application of LPCMCI to a river discharge dataset. A first analysis shows encouraging results (given our understanding of the causal mechanisms). **Non-time series case:** The idea of increasing effect sizes by default conditioning on parents in principle also applies to the non-time series case. We speculate, however, that the gain is most significant in the presence of strong autocorrelation. Moreover, in the non-time series case LPCMCI finds only few default conditions because $i$) PAGs tend to be more unoriented and $ii$) parentships can only be found after having oriented some colliders first (to find parents one first needs some heads '>', which come for free in the time series case). We already cover the non-autocorrelated case that still has time order, see the $a = 0$ point in Fig. 2B as well as all plots for $a = 0$ in the Supplement, which shows comparable performance of SVAR-FCI and LPCMCI. We additionally ran experiments in the true non-time series case and got similar results. **Higher cardinality:** As we state, there is a tradeoff between the positive effect of conditioning on parents and the negative effect of higher cardinality. For the setting that LPCMCI is designed for, autocorrelated time series, our experiments show a significant performance gain (excluding the discrete case for now). In other settings, e.g. the non-time series case, the relative effect is less clear. LPCMCI's conditioning sets consist of two parts: The standard PC-like set plus the default conditions (known parents). The cardinality constraint mentioned in L275-277 $i$) only restricts the former part, $ii$) is used only in the last phase of LPCMCI (pseudocode line 6), $iii$) applies to SVAR-FCI too, and $iv$) is used to limit excessive runtime (mostly needed for SVAR-FCI). In the continuous case, loosing $\mathcal{O}(1)$ degrees of freedom by default conditions is negligible to, e.g., $\mathcal{O}(100)$ sample sizes. While we did not implement a constraint on the number of default conditions, this would indeed be a good idea as it would allow to analyze the effect of higher cardinality and might be relevant for the discrete case. **Discrete variables:** Fair point. LPCMCI in principle also works with discrete variables as it can utilize any CI test, but evaluation is needed. While preliminary experiments did not show significant differences between the methods, we will run more experiments and accordingly extend the camera-ready version. The range of applicability of LPCMCI will remain broad in any case. In climate science applications, e.g., there usually are only few discrete variables, if any. **Residualization:** The question is whether instead of conditioning on parents one might use a residualization procedure in data preprocessing. We ran two tests. 1) Fit independent AR(1) models and run SVAR-FCI on the residuals. 2) Instead of AR(1) use GP regression as proposed in Flaxman et al., 2016 (using sklearn with RBF kernel and $\alpha = 1$). In both cases adjacency TPR and orientation recall increase but are still lower than for LPCMCI, whereas adjacency FPR increases and orientation precision drops. Among the two, AR(1) performed better. Generally, we are not sure what the ground truth MAG / PAG should be after residualization. Perhaps they should not contain auto-links. This seems to require a substantially different theory.

Other questions and comments will all be addressed by further explanations in the camera-ready version, here our brief answers. **Do we compare to genuine FCI or SVAR-FCI?** To SVAR-FCI, as stated in L103f. **Relation of Theorem 1 to higher recall:** For a single CI test with null $I(X, Y|Z) = 0$ and alternative $I(X, Y|Z) > 0$ the effect size is the value of $I(X, Y|Z)$ in the true (unknown) distribution. For $X$ and $Y$ adjacent, effect size $I(X, Y|Z) > 0$. The larger this true value, the higher the probability of its sample value lying in the test's rejection region and hence of correctly retaining the edge (thus higher recall). Recall is influenced both by effect size and by the cardinality of $Z$, with the details depending on the particular test statistic. **How are parents determined?** LPCMCI alternates between performing CI tests and applying orientation rules, the latter of which may identify some parentships that are then used as default conditions in the next iteration of CI tests. See also L230-236. **Relation to PCMCI:** Our work borrows, and by means of Theorem 1 formalizes, PCMCI's intuition that effect size increases by default conditioning on parents. PCMCI does use default conditions, but it tries to limit their number. In the causally insufficient setting of LPCMCI, bidirected edges can point into the past. To ensure that no m-separations are destroyed, all default conditions must be ancestors of $X$ or $Y$ (though only parents are used to not make cardinality unnecessarily large). This requires orienting edges before having found a final skeleton, which in turn requires our new graphical theory in Secs. 3.2 and 3.3. **Not assuming orientation-faithfulness:** LPCMCI orients colliders not with the potentially overly restrictive 'conservative rule' but with a variant of the 'majority rule' (Colombo and Maathuis, 2014). It also marks conflicts when contradicting orientations are proposed. We assume full faithfulness to prove soundness and do not attempt to discover violations of orientation-faithfulness (we are not aware of such work in the causally insufficient case). **Use of known parents in Lee and Honavar 2017:** We will add a citation. **Small drop of precision drop from $T = 500$ to $T = 1000$:** We found a consistent slight decrease in precision only for contemporaneous links and strong autocorrelation, whereas for lagged links precision sometimes even slightly increases (see Fig. 12 bottom right). Since cardinality increases for both type of links, we do not see an easy explanation. **'Stationarity is enforced':** Whenever an edge is removed (oriented), all equivalent time shifted edges are removed too (oriented in the same way). **Taking into account background knowledge about parentship:** Yes, exactly! We plan to implement this in a future version of LPCMCI.

[Meta-Review · NeurIPS 2020]

The reviewers appreciated the theoretical and empirical improvements, and notably the clarity of the authors in stating the method and its uses: the method was not overstated and did not make outrageous claims, things that could be more common in current ML literature. The rebuttal improved the reviewer scores. They agreed on the importance of a real data example, and if the authors are able to incorporate this among their other suggestions it will make the paper stronger. I vote to accept.